# WEAK TO STRONG GENERALIZATION FOR LARGE LANGUAGE MODELS WITH MULTI-CAPABILITIES

**Yucheng Zhou[1], Jianbing Shen[1]\*, Yu Cheng[2]\***
[1]SKL-IOTSC, CIS, University of Macau, [2]The Chinese University of Hong Kong
`yucheng.zhou@connect.um.edu.mo`
`jianbingshen@um.edu.mo, chengyu@cse.cuhk.edu.hk`

## ABSTRACT

As large language models (LLMs) grow in sophistication, some of their capabilities surpass human abilities, making it essential to ensure their alignment with human values and intentions, i.e., Superalignment. This superalignment challenge is particularly critical for complex tasks, as annotations provided by humans, as weak supervisors, may be overly simplistic, incomplete, or incorrect. Previous work has demonstrated the potential of training a strong model using the weak dataset generated by a weak model as weak supervision. However, these studies have been limited to a single capability. In this work, we conduct extensive experiments to investigate weak to strong generalization for LLMs with multi-capabilities. The experiments reveal that different capabilities tend to remain relatively independent in this generalization, and the effectiveness of weak supervision is significantly impacted by the quality and diversity of the weak datasets. Moreover, the self-bootstrapping of the strong model leads to performance degradation due to its overconfidence and the limited diversity of its generated dataset. To address these issues, we proposed a novel training framework using reward models to select valuable data, thereby providing weak supervision for strong model training. In addition, we propose a two-stage training method on both weak and selected datasets to train the strong model. Experimental results demonstrate our method significantly improves the weak to strong generalization with multi-capabilities.

## 1 INTRODUCTION

With large language models (LLMs) continuing to grow in strength and sophistication, some of their capabilities surpass human abilities, e.g., text summarization (Pu et al., 2023), predicting neuroscience results (Luo et al., 2024), etc. Consequently, Burns et al. (2023) introduce the concept of "superalignment", which seeks to guarantee that these superhuman models' capabilities remain aligned with human intentions and values when understanding and executing tasks.

As LLMs become increasingly powerful, they can handle more complex tasks, often exhibiting reasoning abilities that exceed those of humans. This progression underscores the challenge of providing sufficient labeled data for training LLMs, especially for intricate tasks. For such complex tasks, humans may only be able to offer simple, incomplete, or even erroneous annotations. Despite this, LLMs are expected to understand human intent to solve these tasks effectively. Some works (Burns et al., 2023; Gambashidze et al., 2024) investigate the weak to strong generalization setting, which involves training a strong model using data generated by a weak model. In this setting, the weak model is analogous to humans, providing weak supervision to guide the training of the strong model, similar to how humans teach a superhuman model. However, previous work (Burns et al., 2023; Liu & Alahi, 2024; Gambashidze et al., 2024) has only demonstrated weak to strong generalization in single capabilities. A strong model typically necessitates a range of capabilities, and a single weak model

---

\*Corresponding Author. This work was supported by the National Natural Science Foundation of China (No. 624B2002), the Science and Technology Development Fund of Macau SAR (FDCT) under grants 0102/2023/RIA2 and 001/2024/SKL, the Jiangyin Hi-tech Industrial Development Zone under the Taihu Innovation Scheme (EF2025-00003-SKL-IOTSC), and the University of Macau SRG2022-00023-IOTSC grant.

may lack the capabilities required. Thus, weak-to-strong generalization across multiple capabilities is essential, enabling strong model to acquire diverse abilities from different weak models.

In this work, we conduct extensive experiments to investigate the multi-capabilities weak to strong generalization. Firstly, we validate the weak to strong generalization for LLMs with multiple capabilities. Subsequently, we analyze the interactions between these capabilities during the generalization process, aiming to identify the factors influencing performance. Preliminary experiments reveal that the different capabilities exhibit a tendency to remain relatively independent, irrespective of the strength of the generalization. Moreover, the quality of the generated data significantly impacts the effectiveness of weak supervision. Upon delving deeper into the weak to strong generalization, we discover that strong models do not strictly adhere to weak supervision, i.e., weak models. Further analysis indicates that strong models tend to exhibit overconfidence in certain knowledge in a capability, which also leads to a degradation in performance through its self-bootstrapping. The analysis demonstrates that data generated by weak models tends to be more diverse for the strong model, which is more beneficial for the weak to strong generalization.

To improve weak to strong generalization with multi-capabilities, we propose a novel training framework that employs reward models to select valuable data as weak supervision for strong model training. Firstly, generating weak datasets by weak models serves as weak supervision. Subsequently, the strong model trained on weak datasets relabels the weak datasets to obtain strong datasets. Furthermore, according to previous findings of inconsistency between weak models and strong models, we divide the weak datasets and strong datasets into three parts based on consistency: consistent datasets, inconsistent weak datasets, and inconsistent strong datasets. In previous findings, weak datasets exhibit greater diversity relative to the strong model and are better beneficial for weak to strong generalization. Consequently, we train a reward model to identify correct datasets that differ in distribution from the strong datasets. In addition, we then propose a two-stage training method, enabling the strong model to learn from both the weak dataset and the selected datasets, thereby achieving superior multi-capabilities weak to strong generalization.

In this study, our contributions are as follows:

- We conducted extensive experiments to investigate weak to strong generalization with multi-capabilities. The analysis results provide insights, including the relative independence of different capabilities and weak datasets more diverse compared with strong datasets.
- We propose a novel training framework for strong model training, consisting of a reward model for selecting valuable weak supervision data, and two-stage training for the strong model on weak datasets and selected datasets.
- Our method significantly enhances weak to strong generalization for LLMs with multi-capabilities. Further analysis demonstrates that the reward model can select more accurate data that is also more divergent from the data distributions generated by strong models.

## 2  BACKGROUND AND NOTATION

Weak to strong generalization in large language models refers to a weak model guiding a strong model training, an analogy setting of human-guided superhuman model training (Burns et al., 2023). Specifically, the weak model generates a weak dataset to provide weak supervision for the strong model's training. However, the data generated by the weak model may contain errors and be incomplete. Consequently, the strong model needs to infer the correct task intentions from the imperfect guidance provided by the weak model. Following Burns et al. (2023), the weak to strong generalization involves two different training sets with a single capability $i$, namely $\mathcal{D}_i^a$ and $\mathcal{D}_i^b$. In this setting, the dataset $\mathcal{D}_i^a$ is annotated by humans, while $\mathcal{D}_i^b$ is an unannotated dataset.

Firstly, the weak model $M_i^{(w)}$ is trained on the human-annotated dataset $\mathcal{D}_i^a$, resulting in the trained weak model $\bar{M}_i^{(w)}$. This ensures that the model acquires a certain capability $i$, similar to human understanding. This training process can be expressed as follows:

$$\min_{\theta_w} \sum_{(x,y)\in\mathcal{D}_i^a} \mathcal{L}(y, f_{\theta_{w,i}}(x)) \tag{1}$$

where $\mathcal{L}$ denotes the loss function, $\theta_{w,i}$ represents the weak model parameters, and $(x,y)$ are the labeled samples from the dataset $\mathcal{D}_i^a$.

Subsequently, the trained weak model is used to label the unannotated dataset $\mathcal{D}_i^b$, resulting in the dataset $\bar{\mathcal{D}}_i^b$. Since the weak model's capability $i$ may be limited, its annotations on $\bar{\mathcal{D}}_i^b$ could be incomplete and noisy. The annotation process by the weak model can be described as follows:

$$\bar{\mathcal{D}}_i^b = \{(x, \hat{y}) \mid x \in \mathcal{D}_i^b, \hat{y} = f_{\bar{\theta}_{w,i}}(x)\} \tag{2}$$

where $\bar{\theta}_{w,i}$ is the parameters of trained weak model, and label $\hat{y}$ is predicted by trained weak model.

In weak to strong generalization, the strong model $M_i^{(s)}$ learns the capability $i$ by training on the dataset $\bar{\mathcal{D}}_i^b$ annotated by the weak model. The training process for the strong model is as follows:

$$\min_{\theta_s} \sum_{(x, \hat{y}) \in \bar{\mathcal{D}}_i^b} \mathcal{L}(\hat{y}, f_{\theta_{s,i}}(x)) \tag{3}$$

where $\theta_{s,i}$ represents the parameters of the strong model, and $\mathcal{L}$ denotes the loss function for training the strong model. The objective is to minimize $\mathcal{L}$ by adjusting the parameters $\theta_{s,i}$ so that the strong model can effectively learn from the annotations generated by the weak model.

**Related Work.** Recent advancements have been marked by the development of large language models (LLMs), e.g., GPT-4 (OpenAI, 2023), Gemini (Anil et al., 2023), LLaMA (Touvron et al., 2023), Qwen (Bai et al., 2023). Due to some capabilities of LLMs surpassing those of humans (Pu et al., 2023; Luo et al., 2024), a significant focus has been on weak to strong generalization. Studies like (Tong et al., 2024; Li et al., 2024a; Sun et al., 2024) have shown how this approach optimizes capabilities through methods like self-reinforcement and data filtering. In addition, training data selection methods, both efficiency-based (Xie et al., 2023; Zhou et al., 2023) and quality-based (Wettig et al., 2024; Yu et al., 2023)), have been developed to enhance LLM performance by focusing on computational efficiency and data quality. The full version can be found in Appendix B.

## 3 ANALYSIS ON MULTI-CAPABILITIES WEAK TO STRONG GENERALIZATION

### 3.1 MULTI-CAPABILITIES FOR LLM

Numerous studies are currently investigating the various capabilities of LLMs (Chen et al., 2024; Zhao et al., 2024), including mathematics (Li et al., 2024b), temporal reasoning (Zhou et al., 2019), planning (Logeswaran et al., 2022), and more. Unlike previous research that primarily focused on a single ability (Burns et al., 2023), our study aims to explore whether weak to strong generalization can occur across multiple capabilities in LLMs. To explore the models' performance in weak to strong generalization settings, we employ eight datasets that span various skills, such as GSM8K (Cobbe et al., 2021) for mathematical abilities, MC-TACO (Zhou et al., 2019) for temporal reasoning, SCAN (Lake & Baroni, 2018) for planning ability, CREAK (Onoe et al., 2021) for fact-checking and commonsense reasoning ability, ECQA (Aggarwal et al., 2021) for explainable commonsense reasoning, e-SNLI (Camburu et al., 2018) for logical reasoning ability, OpenBookQA (Mihaylov et al., 2018) for fact reasoning, and SciQ (Welbl et al., 2017) for science-related abilities.

### 3.2 CAN TRAINING MODEL WITH MULTI-CAPABILITIES FROM WEAK SUPERVISION?

To investigate whether LLMs can learn multiple capabilities from multiple weak supervision, we train a series of LLMs with different parameter scales (i.e., 0.5B, 1.8B, 4B, 7B) on datasets annotated for these capabilities. The average performance of these LLMs in multiple capabilities is shown in Figure 1 (Left). From the figure, we can observe that larger LLMs, i.e., with more parameters, exhibit better average performance, confirming a positive correlation between model size and capability in these LLMs. Based on the premise that these models, ordered from smallest to largest, exhibit a corresponding increase in capability, we conduct weak to strong generalization experiments on these models. Following

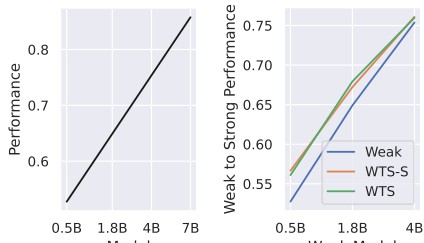

Figure 1: **Left**: Performance of different-scale models on multi-capabilities tasks; **Right**: Performance of strong model with single-capability and multi-capabilities weak to strong generalization.

Burns et al. (2023), we employ weaker models to generate datasets that served as weak supervision for stronger model training, i.e., 0.5B, 1.8B, and 4B models provide weak supervision for 1.8B, 4B, and 7B models, respectively. The results, shown in Figure 1 (Right), reveal that the average performance of the weaker models consistently underperforms compared to the "WTS-S" (averaging performance across models with single-capability weak to strong generalization on different datasets) and "WTS" (averaging performance of model with multi-capabilities weak to strong generalization). The performance between single-capability and multi-capabilities generalization is comparable. In addition, we can observe that the gains from weak to strong generalization are more pronounced when the model size is smaller due to its weaker capability. Further details, i.e., the performance of each capability for Figure 1, can be found in Appendix D.

### 3.3 HOW DOES WEAK MODEL GENERATED DATA IMPACTS WEAK TO STRONG GENERALIZATION WITH MULTI-CAPABILITIES?

To further explore the factors impacting multi-capabilities weak to strong generalization, we train strong models on datasets generated by weak models of varying model sizes. As shown in Figure 2, the quality of data generated by weak models significantly impacts the performance of weak to strong generalization. Higher-quality data generated by weak models with greater model sizes improve the performance of stronger models, highlighting the importance of improving the data quality produced by these weak models. Beyond the direct impact of the generated data, we also analyze the effect of dataset compositions with different capabilities on strong models. In our experiment, we randomly weaken certain capabilities in weak supervision. Specifically, we replace the training data for given capabilities with weak data generated by a smaller-size weak model corresponding to that capability. As shown in Figure 3, we observe that the performance of other capabilities in the strong model is not

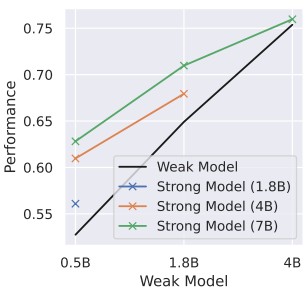

Figure 2: Performance of strong model trained on different weak supervision.

significantly affected. It demonstrates that weak data quality is relatively independent across different capabilities. To further validate this observation, we randomly remove some capabilities to investigate any potential synergistic or inhibitory interactions among capabilities. The results, presented in Figure 4, confirm that removing certain capabilities does not compromise strong model performance in others, supporting the independence of data quality across various capabilities. More results of strong models on each capability for Figure 2, Figure 3, and Figure 4, can be found in Appendix E.

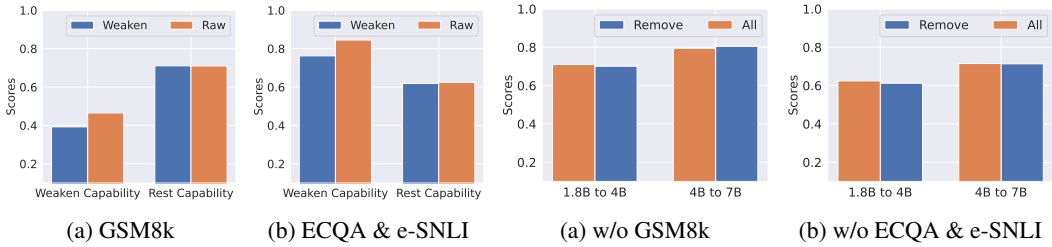

| (a) GSM8k | (b) ECQA & e-SNLI | (a) w/o GSM8k | (b) w/o ECQA & e-SNLI |

Figure 3: Results under weaken some capabilities.  Figure 4: Results under remove some capabilities.

### 3.4 HAVE CONSISTENCY BETWEEN STRONG MODEL AND WEAK SUPERVISION?

To deepen our understanding of the performance improvements exhibited by strong models in weak to strong generalization, we analyze the consistency of prediction results between strong and weak models. Firstly, weak models generate datasets for training strong models, which are then re-annotated these datasets by the trained strong models. As shown in Figure 5, the majority of predictions by the strong and weak models were consistent, denoted as "True Con." and "False Con.". This indicates that strong models predominantly learn from the knowledge provided by weak models. However, it is observed that strong models correctly predict more samples compared to weak models, i.e., "WTS True" versus "Weak True". This suggests that the essence of weak to strong generalization is not

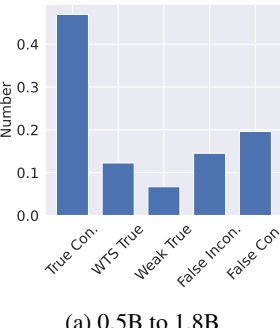 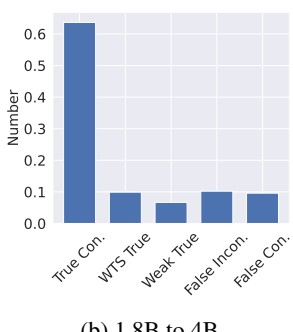 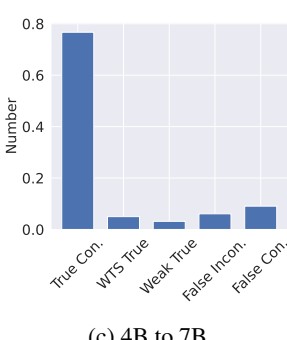

| (a) 0.5B to 1.8B | (b) 1.8B to 4B | (c) 4B to 7B |

Figure 5: Prediction results of strong model (WTS) and weak supervision (Weak). "True Con." and "False Con." indicate cases where both models give the same correct or incorrect answers, respectively. "WTS True" means the strong model is correct and the weak model is incorrect, while "Weak True" is the opposite. "False Incon." indicates both models are incorrect but with different answers.

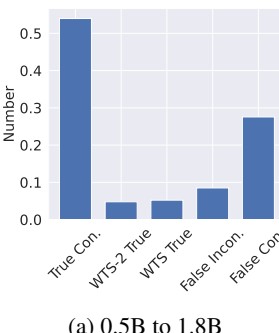 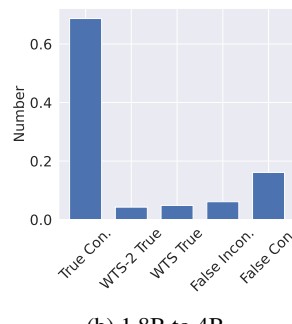 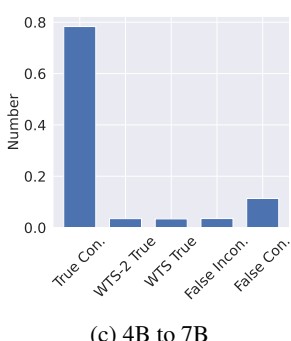

| (a) 0.5B to 1.8B | (b) 1.8B to 4B | (c) 4B to 7B |

Figure 6: Prediction results of strong model (WTS-2) and its supervision from strong model (WTS).

merely the strong model replicating the weak model but rather learning the task's intent and rules from the datasets. Nevertheless, the strong models incorrectly answer "Weak True" samples correctly predicted by the weak models, highlighting a potential overconfidence issue in strong models that prevent them from learning some correct samples identified by the weak models. The details of each capability consistency between strong model and weak supervision can be found in Appendix F.

### 3.5 CAN IMPROVE STRONG MODEL WITH SELF-BOOTSTRAPPING?

Based on the above analysis, we can observe that a strong model trained on datasets generated by weak models can achieve better performance. This raises a new question: "Can a strong model continuously improve itself through self-bootstrapping?". To explore its answer, we relabel the datasets using the trained strong model (WTS) and then trained a second version of the strong model (WTS-2) on the relabeled datasets. We compared the predictions of the trained strong model and the second version for consistency. As shown in Figure 6, we found that the second version correctly labeled fewer samples than the original strong model, i.e., "WTS-2 True" vs. "WTS True". This indicates that it fails to enhance the performance of weak to strong generalization. As mentioned in Sec. 3.4, the strong model tends to be overconfident, leading to data collapse into a particular distribution during self-bootstrapping. Similar conclusions have been observed in other studies (Alemohammad et al., 2023; Shumailov et al., 2023; Seddik et al., 2024). More details of each capability can be found in Appendix G.

### 3.6 HOW THE WEAK DATASETS IMPACT THE WEAK TO STRONG GENERALIZATION PERFORMANCE?

Given that self-bootstrapping the strong model does not yield improved results, we focus on how datasets generated by the weak model impact the weak to strong generalization. Firstly, we train the

strong model on the clean weak datasets that include only the correct sample in datasets generated by the weak model. Subsequently, we introduce varying proportions of noise (i.e., incorrect samples from the same weak dataset) to the clean datasets for strong model training. As shown in Figure 7 (Left), the strong model performs best on clean datasets, which indicates that training on accurate weak data yields better results, emphasizing the importance of filtering out erroneous samples in weak datasets. In addition, as shown in Figure 7 (Right), we use the strong model trained on the relabeled data from the strong model, termed "Strong Data", to train a second version of the strong model. We ob-

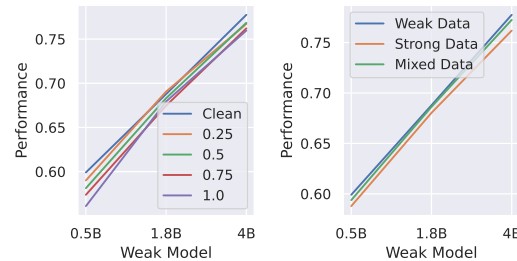

Figure 7: **Left**: Performance of strong model trained on clean samples and different proportions of noise samples of weak datasets; **Right**: Performance of strong model trained on weak, strong, and combination datasets.

serve that its performance is inferior to the strong model trained on "Weak Data", further confirming the failure of model self-bootstrapping. Moreover, to mitigate the distributional collapse caused by the overconfidence of the strong model, we combine the data labeled by both the weak model and the trained strong model for training the strong model, "Mixed Data" in the figure. Although this approach improves performance compared to "Strong Data", it still does not surpass the performance achieved with "Weak Data". This underscores the significance of weak data for training strong models. The performance on each capability can be found in Appendix H.

## 4 MULTI-CAPABILITIES WEAK TO STRONG GENERALIZATION WITH REWARD MODELS

As mentioned before, the significance of weak datasets is shown in Figure 7, which shows that clean weak datasets can provide better supervision for the strong model. A key factor in improving the performance of weak to strong generalization is the accuracy of the weak datasets. Moreover, weak datasets provide greater gain than strong datasets for the strong model. This difference arises from the data distributions of weak datasets and strong datasets. Since strong datasets align more closely with the internal knowledge of the strong model, this knowledge can be more easily activated when using weak datasets to train a strong model. Self-bootstrapping a strong model on its generated data gradually increases the model's confidence in its internal knowledge, leading to potential collapse and overconfidence during prediction. In contrast, weak datasets present a more diverse data distribution for the strong model. Our method improves the strong model by increasing the accuracy of weak datasets and selecting more diverse weak data for the strong model.

### 4.1 THE GOAL OF WEAK DATA SELECTION

As shown in Sec. 2, we first use the labeled dataset $\mathcal{D}_i^a$ to train a weak model $M_i^{(w)}$ and then leverage the trained weak model $\bar{M}_i^{(w)}$ to predict the answer for the unlabeled dataset $\mathcal{D}_i^b$ to obtain labeled weak dataset $\bar{\mathcal{D}}_i^b$. Subsequently, we train a strong model $M_i^{(s)}$ with the weak dataset $\bar{\mathcal{D}}_i^b$. The trained strong model $\bar{M}_i^{(s)}$ is employed to relabel the dataset $\mathcal{D}_i^b$ to obtain the strong dataset $\hat{\mathcal{D}}_i^b$.

Let $P_{\bar{\mathcal{D}}_i^b}(x)$ and $P_{\hat{\mathcal{D}}_i^b}(x)$ represent the data distributions of datasets $\bar{\mathcal{D}}_i^b$ and $\hat{\mathcal{D}}_i^b$, respectively. Our goal is to identify a subset $\widetilde{\mathcal{D}}_i^b \subset \bar{\mathcal{D}}_i^b$ such that the accuracy of samples in $\widetilde{\mathcal{D}}_i^b$ is high and the distribution $P_{\widetilde{\mathcal{D}}_i^b}(x)$ differs significantly from $P_{\hat{\mathcal{D}}_i^b}(x)$, i.e.,

$$\max_{\widetilde{\mathcal{D}}_i^b} \quad \mathbb{E}_{x \sim \widetilde{\mathcal{D}}_i^b}[\text{Accuracy}(x)] \quad \text{subject to} \quad \mathbb{D}_{\text{KL}}(P_{\widetilde{\mathcal{D}}_i^b} \| P_{\hat{\mathcal{D}}_i^b}) \text{ is high.} \quad (4)$$

where $\mathbb{D}_{\text{KL}}$ denotes the Kullback-Leibler divergence between the distributions $P_{\widetilde{\mathcal{D}}_i^b(x)}$ and $P_{\hat{\mathcal{D}}(x)}$, and the divergence can be defined as:

$$\mathbb{D}_{\text{KL}}(P_{\widetilde{\mathcal{D}}_i^b} \| P_{\hat{\mathcal{D}}_i^b}) = \sum_x P_{\widetilde{\mathcal{D}}_i^b}(x) \log \frac{P_{\widetilde{\mathcal{D}}_i^b}(x)}{P_{\hat{\mathcal{D}}_i^b}(x)} \quad (5)$$

This measure ensures that the selected subset $\widetilde{\mathcal{D}}_i^b$ not only maintains high accuracy but also introduces sufficient diversity to prevent a collapse of the strong model.

## 4.2 Training Reward Models

To obtain a subset $\widetilde{\mathcal{D}}_i^b$ for strong model training, we train a reward model to differentiate between data distributions. Following Figure 5, we first divide the data into three parts: consistent dataset $\mathcal{D}_i^c$ from weak dataset $\bar{\mathcal{D}}_i^b$ and strong dataset $\hat{\mathcal{D}}_i^b$, and inconsistent data, with $\bar{\mathcal{D}}_i^c$ representing inconsistent data in $\bar{\mathcal{D}}_i^b$ and $\hat{\mathcal{D}}_i^c$ representing inconsistent data in $\hat{\mathcal{D}}_i^b$. We define the consistent and inconsistent datasets as follows:

$$\mathcal{D}_i^c = \{x \mid x \in \bar{\mathcal{D}}_i^b \cap x \in \hat{\mathcal{D}}_i^b\}$$
$$\bar{\mathcal{D}}_i^c = \{x \mid x \in \bar{\mathcal{D}}_i^b \setminus \hat{\mathcal{D}}_i^b\}$$
$$\hat{\mathcal{D}}_i^c = \{x \mid x \in \hat{\mathcal{D}}_i^b \setminus \bar{\mathcal{D}}_i^b\} \tag{6}$$

In the setting of weak to strong generalization, we do not know which samples in $\bar{\mathcal{D}}_i^b$ and $\hat{\mathcal{D}}_i^b$ are correct. Therefore, we choose $\mathcal{D}_i^c$ as positive samples because they include data where both the weak model and the strong model agree, indicating higher confidence (as shown in Figure 5). In contrast, we select $\hat{\mathcal{D}}_i^c$ as negative samples due to their low confidence. Additionally, this data includes "WTS True" samples from Figure 5, which we know from Sec. 3.5 to be less diverse from strong model. We aim to ensure that the distribution of the training data has high accuracy and differs more from $\hat{\mathcal{D}}_i^c$ to prevent collapse. The objective function for the reward model can be defined as:

$$\max_{\boldsymbol{R}} \quad \mathbb{E}_{x \sim \mathcal{D}_i^c}[\boldsymbol{R}(x)] - \mathbb{E}_{x \sim \hat{\mathcal{D}}_i^c}[\boldsymbol{R}(x)] \tag{7}$$

where $\boldsymbol{R}(x)$ denotes the reward assigned to a sample $x$ by the reward model $\boldsymbol{R}$. This ensures that the reward model favors data points where the weak model's predictions are consistent with the strong model's predictions (i.e., higher accuracy than the inconsistent part) while penalizing inconsistent ones in the strong dataset. The proof of the reward model can be found in Appendix C.

## 4.3 Reward Model for Data Selection

To obtain better datasets from strong model training, we employ the trained reward model $\boldsymbol{R}$ to filter the weak dataset from each capability to collect selected dataset $\ddot{\mathcal{D}}_i^b$. The reward model distinguishes each sample $x$ from weak datasets based on its accuracy and its contribution to diversity. Specifically, the sample, classified as correct by the reward model, exhibits significant divergence from the strong model's data distribution in the dataset $\ddot{\mathcal{D}}_i^b$. In the weak dataset $\bar{\mathcal{D}}_i^b$, each sample is evaluated by the reward model. The reward model, $R(x)$, is a binary classifier that determines whether a sample is beneficial for training the strong model by considering both accuracy and diversity. Samples are filtered based on the reward model classification as follows:

$$\ddot{\mathcal{D}}_i^b = \{x \mid R(x) = 1\}, x \in \bar{\mathcal{D}}_i^b \tag{8}$$

The selected samples have higher accuracy and introduce sufficient diversity for strong model training.

## 4.4 Train Strong Model with Selected Data

The strong model $\boldsymbol{M}^{(s)}$ is trained in two stages to leverage the benefits of both the weak datasets $\bar{\mathcal{D}}^b = \{\bar{\mathcal{D}}_0^b, \cdots, \bar{\mathcal{D}}_N^b\}$ and selected weak datasets $\ddot{\mathcal{D}}^b = \{\ddot{\mathcal{D}}_0^b, \cdots, \ddot{\mathcal{D}}_N^b\}$, where $N$ is the number of capabilities. Firstly, the strong model is trained using the weak datasets $\bar{\mathcal{D}}^b$. This initial training phase serves as a warm-up, allowing the model to learn from a broader range of weak data, even though it may not be entirely accurate or comprehensive.

$$\boldsymbol{M}_i^{(s)} \leftarrow \text{Warm-up}(\boldsymbol{M}_i^{(s)}, \bar{\mathcal{D}}^b), \tag{9}$$

After the warm-up phase, the model is trained using the selected weak dataset $\ddot{\mathcal{D}}^b$. This phase focuses on the higher accuracy, and diverse samples selected by the reward model, ensuring that the strong model refines its predictions and avoids collapse.

$$\bar{\boldsymbol{M}}_i^{(s)} \leftarrow \text{Train}(\boldsymbol{M}_i^{(s)}, \ddot{\mathcal{D}}^b) \tag{10}$$

By employing this two-stage training approach, the strong model benefits from an initial broad exposure to weak data, followed by focused training on higher accuracy and diverse samples. This method enhances the model's generalization capabilities, leveraging the advantages of weak data to improve overall performance while mitigating risks of overconfidence and collapse.

## 5 EXPERIMENTS

### 5.1 EXPERIMENTAL SETUPS

**Multi-Capabilities Task.** For the multi-capabilities task, we utilize a range of datasets encompassing various capabilities to comprehensively evaluate the models' performance under weak to strong generalization settings. The tasks were selected to cover diverse capabilities, such as reasoning, comprehension, robot planning, math skills, etc. Details of the datasets are provided in Appendix A. Following Burns et al. (2023), each capability's dataset is split into a labeled training set, an unlabeled training set, and a test set to simulate the weak to strong generalization scenario. The weak models were trained on the annotated datasets, and their prediction on the unannotated datasets served as the weak supervised dataset for training the strong models.

**Experimental Details.** In the experiments, we utilized a series of Qwen-1.5 models (Bai et al., 2023) with varying parameters, specifically 0.5B, 1.8B, 4B, and 7B. The reward models are initialized from the strong model, i.e., Qwen-1.5, and they maintain the same parameters. To ensure a fair comparison, we followed the experimental setup from Burns et al. (2023), conducting all experiments with 2 epochs and a batch size of 40. The optimizer used was Adam (Kingma & Ba, 2015) with a learning rate of 1e-5. Weight decay was set at 0.01, and a cosine learning rate decay strategy was employed. During inference, the models utilized a greedy decoding strategy. The performance refers to accuracy, where a correct prediction exactly matches the ground truth answer. For weak to strong generalization, we use the performance gap recovered (PGR) metric (Burns et al., 2023) to measure the weak to strong generalization performance. All experiments are conducted on NVIDIA A100 80G GPUs.

### 5.2 RESULTS AND DISCUSSION

As shown in Figure 8, we present the experimental results on the weak to strong generalization for LLMs with multi-capabilities. Our method, denoted as "Ours", is compared against the multi-capabilities weak to strong generalization approach, i.e., "WTS", and the performance of strong models trained on datasets $\mathcal{D}_i^b$ with human annotations is used as a performance ceiling. The average performance of the models is plotted against the size of the strong models (1.8B, 4B, and 7B parameters) in Figure 8. In addition, we show the performance gap recovered of our method and baseline, i.e., "WTS", in Figure 9. The results of each capability can be found in Appendix I.

**Analysis of Performance.** As shown in Figure 8, the "WTS" shows a gradual improvement in performance as the size of the strong model increases from 1.8B to 7B parameters. However, its relatively modest performance improvement indicates that weak to strong generalization benefits from strong model capacities but still faces limitations. Our method, "Ours", significantly outperforms the "WTS" across all strong model sizes. For the same weak model, the performance gap between "Ours" and "WTS" increases as the strong model size increases, suggesting that our approach scales more effectively with stronger models. The performance of the strong model trained on clean datasets is a performance ceiling. While our method does not completely bridge the gap to the performance ceiling, it achieves a substantially closer performance than the "WTS" method, especially for the models with larger sizes. Our method exhibits a notable advantage, indicating its effectiveness even with models with smaller sizes or when the model faces relatively harder tasks. For the 7B model, our method achieves performance comparable

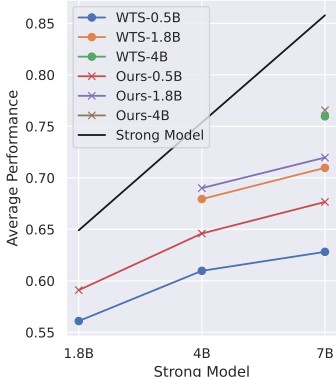

Figure 8: Performance of weak to strong generalization. In "Ours-xB" and "WTS-xB", the "x" indicates the size of the weak model used.

to the "WTS" model. This is because the model already performs at a high level across various capabilities, leaving little room for improvement. In addition, the abundance of correct samples means the reward model's data filtering impact is likely diminished.

**Analysis of Performance Gap Recovered.** In Figure 9, we analyze the performance gap recovered of the baseline "WTS" and our method. This figure shows how much of the performance gap recovered between various methods and the performance ceiling as the model size increases. The "WTS-0.5B", the weak model of 0.5B model, recovers approximately 30% of the performance, while our method recovers around 50%. This indicates that our method is more effective at recovering performance, even with weaker models guiding the strong model. As the model size increases to 7B, the performance gap recovered for both methods slightly decreases. However, our method still outperforms the "WTS" approach. This reduction in recovery at larger model sizes is due to the increased difficulty in further improving already high-performing models and the diminishing impact of reward model filtering as the proportion of correct samples increases.

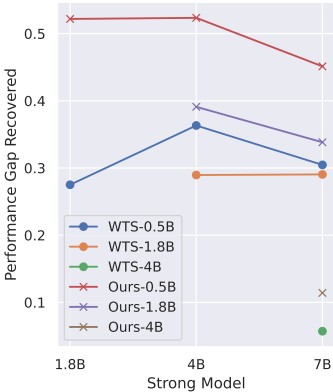

Figure 9: Performance gap recovered of weak to strong generalization.

## 5.3 ANALYSIS

**Deep Dive into Weak Data.** To validate the effectiveness of using weak datasets $\bar{\mathcal{D}}^b$ in the first stage of training, we compared the performance of models trained with a first-stage weak dataset $\bar{\mathcal{D}}^b$ against those trained directly on the selected weak dataset $\ddot{\mathcal{D}}^b$ without a first-stage training. The comparison spanned models of varying sizes, including 0.5B, 1.8B, and 4B, with corresponding strong models at 1.8B, 4B, and 7B. As shown in Figure 10, models trained using weak datasets $\bar{\mathcal{D}}^b$ in the first stage consistently outperform those that skip this stage and train directly on $\ddot{\mathcal{D}}^b$. This demonstrates that even though weak datasets $\bar{\mathcal{D}}^b$ may not be entirely accurate or comprehensive, they still provide a broader range of weak data that benefits model training.

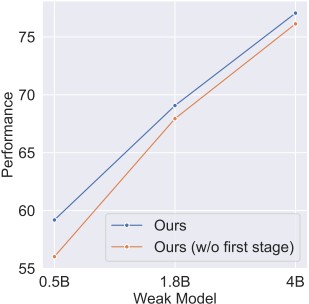

Figure 10: Performance of model with and without first stage training on weak datasets $\bar{\mathcal{D}}^b$.

**Impact of Reward Model.** To analyze how our method achieves performance gains through the selection of weak data, we further examined the accuracy of the selected dataset. As shown in Figure 11, we present the accuracy of datasets provided by different models, including the weak model, strong model, and our reward model. The reward model consistently outperforms both the weak and strong models across different weak model sizes (0.5B, 1.8B, and 4B). Specifically, the accuracy of the reward model increases significantly as the weak model size grows. This indicates that the reward model and the weak model are correlated, as the weak model provides positive training samples for the reward model, as mentioned in Sec. 4.2. As the capability of the weak model improves, it provides more accurate positive training samples. However, the performance gains from the reward model diminish as the weak model's performance improves because the task becomes more challenging. In addition, in Figure 11, we also show the weak dataset retention rate of our reward model, represented by the gray bars. The figure shows as the model size increases, the reward model not only improves accuracy but also retains a higher proportion of useful data.

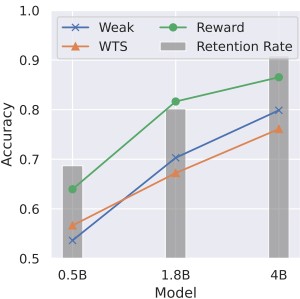

Figure 11: Accuracy of datasets from reward model, weak model, and strong model.

**Deep Dive into Reward Model.** In Figure 11, we have demonstrated that the accuracy of the data selected by the reward model is higher than that of the original weak data. Thus, we aim to observe whether the distribution of this data differs significantly from the distribution of the strong data, specifically whether the reward model can select more data labeled as "Weak True". Based

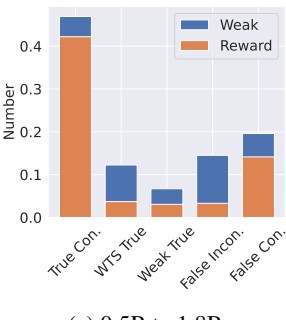
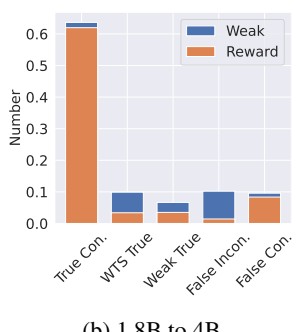
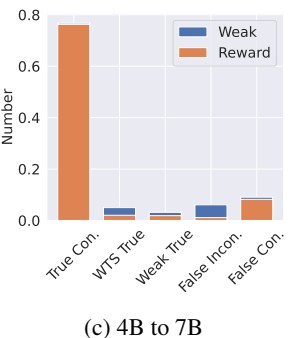

(a) 0.5B to 1.8B          (b) 1.8B to 4B          (c) 4B to 7B

Figure 12: Performance of reward model (Reward) on (in)consistent part between weak model (Weak) and strong model.

on Figure 5, we conduct a distribution analysis of the data selected by the reward model, with the results shown in Figure 12. Firstly, we observe that the reward model retains a higher proportion of "Weak True" data among the inconsistency between the weak model and strong model predictions, i.e., "Strong True", "Weak True", and "False Incon". It demonstrates that the reward model not only improves accuracy but also maintains a data distribution distinct from the strong model's predictions. Furthermore, as the weak model with greater size, the proportion of "True Con." increases, leading to a slight improvement by the reward model, as shown in Figure 8. More details of the reward model can be found in Appendix J.

**Analysis Across Model Scales.** To further analyze whether our approach remains effective with larger models, approaching more realistic settings, we expanded the range of weak models to include 0.5B, 1.8B, 4B, 7B and 32B, with their corresponding strong models set at 1.8B, 4B, 7B, 32B and 72B, respectively. As shown in Figure 13, our method consistently outperforms the "WTS" approach across all strong model sizes. Figure 14 shows that our method consistently maintains a performance recovery rate of over 30%, even as model sizes grow, particularly for weak models of 7B and 32B and strong models of 32B and 72B. Moreover, the results demonstrate that as the scale difference between weak and strong models grows (e.g., weak models from 4B to 32B with their corresponding strong models from 7B to 72B), the performance recovery rate becomes even more significant.

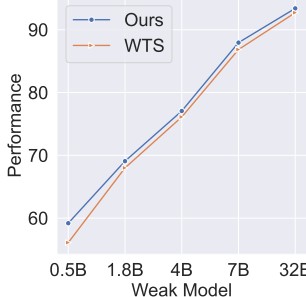

Figure 13: Performance of weak to strong generalization under larger size models.

# 6 CONCLUSION

In this study, we have explored the concept of multi-capabilities weak to strong generalization in large language models (LLMs). Our research validates that LLMs can indeed generalize from weak supervision across multiple capabilities, shedding light on the relative independence of these capabilities during the generalization process. Through extensive experiments, we confirmed that the diversity and quality of data generated by weak models are crucial factors influencing the effectiveness of weak to strong generalization. Specifically, weak models tend to produce more diverse datasets for the strong models, counteracting the performance degradation caused by the overconfidence of strong models. We introduced a novel training framework that incorporates reward models to select high-quality weak supervision data, facilitating superior generalization for strong

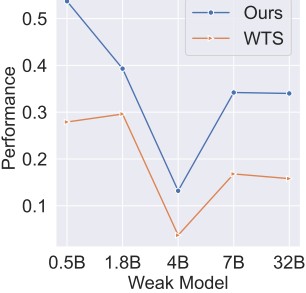

Figure 14: Performance gap recovery of weak to strong generalization under larger size models.

models. Our proposed two-stage training method enables the strong model to learn effectively from both weak and selected datasets, addressing the inconsistencies between weak and strong datasets.

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

# A DETAILS OF DATASETS ON DIFFERENT CAPABILITIES

In this section, we provide a comprehensive overview of the datasets utilized in our study, which are designed to train and evaluate various capabilities of LLMs. Table 1 summarizes the datasets employed, along with their sources and descriptions of the tasks.

Table 1: Datasets on multi-capabilities tasks

| Dataset | Original Source | Description |
| --- | --- | --- |
| CREAK | Onoe et al. (2021) | Fact-Checking and Commonsense Reasoning |
| ECQA | Aggarwal et al. (2021) | Explainable Commonsense Reasoning |
| e-SNLI | Camburu et al. (2018) | Logical Reasoning |
| GSM8K | Cobbe et al. (2021) | Mathematical Problem Solving |
| MC-TACO | Zhou et al. (2019) | Temporal Reasoning |
| OpenBookQA | Mihaylov et al. (2018) | Fact Questions |
| SCAN | Lake & Baroni (2018) | Robot Planning |
| SciQ | Welbl et al. (2017) | Science Question Answering |

# B RELATED WORK

## B.1 LARGE LANGUAGE MODELS

Recent advancements in AI have been driven by the development of large language models (LLMs). These models exhibit remarkable capabilities not only in language reasoning but also in visual understanding (Zhu et al., 2024; Zhou et al., 2024a) and have even demonstrated impressive results across a variety of applications (Zhou et al., 2025; 2024b). These models can be categorized into closed-source and open-source LLMs. Closed-source models such as GPT-4 (OpenAI, 2023), Gemini (Anil et al., 2023), and Claude 3 (Anthropic, 2024) excel in text and visual processing, reasoning, and coding tasks, demonstrating near-human performance on various benchmarks due to significant resource investments but with restricted access. In contrast, open-source models like Mistral 7B (Jiang et al., 2023), LLaMA (Touvron et al., 2023), LLaMA-3 (AI@Meta, 2024) and Qwen (Bai et al., 2023) promote broader research and innovation. Mistral 7B, with 7.3 billion parameters, and LLaMA, ranging from 700 million to 65 billion parameters, provide competitive performance on public datasets, enhancing accessibility and community contributions. Qwen encompasses a variety of models ranging from 0.5 billion to 110 billion parameters, and its range makes it suitable for research on model size and capability, from weak to strong generalization (Burns et al., 2023).

## B.2 WEAK TO STRONG GENERALIZATION

weak to strong generalization leverages weaker models to supervise stronger ones, improving their performance significantly (Burns et al., 2023; Guo et al., 2024). Several studies in weak to strong generalization have focused on optimizing the capabilities of large language models (LLMs) using weak supervision. For instance, Tong et al. (2024) introduces a self-reinforcement approach, starting with supervised fine-tuning (SFT) on a small set of annotated questions. The model then iteratively improves by learning from the differences in responses between the SFT and the un-fine-tuned model on unlabeled questions. In mathematical reasoning, (Zhang et al., 2024) discuss the "AutoDS" method and "AutoMathText" dataset, and uses foundational language models as zero-shot validators for autonomous data selection, significantly improving performance on mathematical tasks. In addition, Li et al. (2024a) investigates using smaller, weaker models for data filtering to fine-tune larger, stronger models in instruction tuning. Despite the performance gap, there is a high consistency between weak and strong models in perceiving instruction difficulty and data selection outcomes.

Other studies have applied weak to strong generalization to specific tasks. For example, Gambashidze et al. (2024) introduces a framework for 3D object detection that addresses sparsity and occlusion issues in LiDAR-based detection. In scalable alignment, Sun et al. (2024) proposes an easy-to-hard generalization method. This approach trains process supervision reward models on easy problems

and uses them to evaluate policy models on harder problems, resulting in significant performance improvements through re-ranking or reinforcement learning. Moreover, Liu & Alahi (2024) presents a co-supervised learning method that enhances weak to strong generalization by using a group of specialized teachers instead of a single generalist teacher. Furthermore, Ji et al. (2024) introduces Aligner, an efficient alignment paradigm that bypasses the RLHF process by learning the correction residuals between aligned and misaligned answers.

### B.3 TRAINING DATA SELECTION

In recent years, various methods have been proposed to enhance the performance of large language models (LLMs) by selecting high-quality training data. These methods can be broadly categorized into algorithms focusing on the efficiency of data selection and those optimizing for specific quality attributes.

Efficiency-based data selection methods prioritize computational efficiency and scalability. The DSIR framework (Xie et al., 2023) estimates importance weights in a simplified feature space and uses importance resampling to select data efficiently, which can select 100M documents from the entire Pile dataset within 4.5 hours. LIMA (Zhou et al., 2023) and LESS (Xia et al., 2024) improve instruction-following capabilities by fine-tuning models using high-quality instruction-tuning data. In addition, Muennighoff et al. (2023) explores scaling language models under data constraints, proposing a method that considers diminishing returns from repeated tokens and excess parameters to achieve computational optimality.

Quality-based data selection methods focus on selecting data that meets certain quality criteria. QuRating (Wettig et al., 2024) aims to select pre-training data that captures abstract qualities of text perceived by humans, such as writing style, required expertise, factuality, and educational value. Moreover, MetaMath (Yu et al., 2023) focuses on mathematical reasoning by self-bootstrapping new mathematical questions from multiple perspectives without adding extra knowledge, resulting in the MetaMathQA dataset. To address the complexities and sensitivities of reinforcement learning from human feedback (RLHF), Song et al. (2024) introduces PRO, an efficient supervised fine-tuning algorithm that iteratively compares candidate responses to guide LLMs towards optimal responses, aligning with human values.

## C PROOF FOR REWARD MODEL

To prove that training a classifier using the defined positive and negative samples will result in positive samples with higher accuracy and a distribution that is further from the strong dataset, we follow these steps:

### C.1 NOTATION AND ASSUMPTIONS

Let:

- $\bar{\mathcal{D}}_i^b$: Weak dataset.

- $\hat{\mathcal{D}}_i^b$: Strong dataset.

- $\mathcal{D}_i^c$: Consistent dataset where $\mathcal{D}_i^c = \{x \mid x \in \bar{\mathcal{D}}_i^b \cap x \in \hat{\mathcal{D}}_i^b\}$.

- $\bar{\mathcal{D}}_i^c$: Inconsistent data in $\bar{\mathcal{D}}_i^b$ where $\bar{\mathcal{D}}_i^c = \{x \mid x \in \bar{\mathcal{D}}_i^b \setminus \hat{\mathcal{D}}_i^b\}$.

- $\hat{\mathcal{D}}_i^c$: Inconsistent data in $\hat{\mathcal{D}}_i^b$ where $\hat{\mathcal{D}}_i^c = \{x \mid x \in \hat{\mathcal{D}}_i^b \setminus \bar{\mathcal{D}}_i^b\}$.

- $\boldsymbol{R}(x)$: Reward assigned to sample $x$ by the reward model.

### C.2 OBJECTIVE FUNCTION

The objective function for the reward model is given by:

$$\max_{\boldsymbol{R}} \quad \mathbb{E}_{x \sim \mathcal{D}_i^c}[\boldsymbol{R}(x)] - \mathbb{E}_{x \sim \hat{\mathcal{D}}_i^c}[\boldsymbol{R}(x)] \tag{11}$$

## C.3   PROOF

We start by defining the expected accuracy of the weak model $\bar{\mathcal{M}}$ and the strong model $\hat{\mathcal{M}}$ on their respective datasets:

$$\text{Accuracy of } \bar{\mathcal{M}} : A_{\bar{\mathcal{M}}} = \mathbb{E}_{x \sim \bar{\mathcal{D}}_i^b}[\mathbf{1}\{\bar{\mathcal{M}}(x) = y\}] \tag{12}$$

$$\text{Accuracy of } \hat{\mathcal{M}} : A_{\hat{\mathcal{M}}} = \mathbb{E}_{x \sim \hat{\mathcal{D}}_i^b}[\mathbf{1}\{\hat{\mathcal{M}}(x) = y\}] \tag{13}$$

where $y$ is the true label of $x$.

By construction, the consistent dataset $\mathcal{D}_i^c$ includes samples where both models agree, which implies higher confidence:

$$\mathbb{E}_{x \sim \mathcal{D}_i^c}[\mathbf{1}\{\bar{\mathcal{M}}(x) = y\}] \geq \mathbb{E}_{x \sim \bar{\mathcal{D}}_i^b}[\mathbf{1}\{\bar{\mathcal{M}}(x) = y\}] \tag{14}$$

$$\mathbb{E}_{x \sim \mathcal{D}_i^c}[\mathbf{1}\{\hat{\mathcal{M}}(x) = y\}] \geq \mathbb{E}_{x \sim \hat{\mathcal{D}}_i^b}[\mathbf{1}\{\hat{\mathcal{M}}(x) = y\}] \tag{15}$$

Next, consider the inconsistent datasets $\bar{\mathcal{D}}_i^c$ and $\hat{\mathcal{D}}_i^c$. These sets include samples where the weak and strong models disagree, indicating lower confidence and potentially higher noise:

$$\mathbb{E}_{x \sim \bar{\mathcal{D}}_i^c}[\mathbf{1}\{\bar{\mathcal{M}}(x) = y\}] \leq \mathbb{E}_{x \sim \bar{\mathcal{D}}_i^b}[\mathbf{1}\{\bar{\mathcal{M}}(x) = y\}] \tag{16}$$

$$\mathbb{E}_{x \sim \hat{\mathcal{D}}_i^c}[\mathbf{1}\{\hat{\mathcal{M}}(x) = y\}] \leq \mathbb{E}_{x \sim \hat{\mathcal{D}}_i^b}[\mathbf{1}\{\hat{\mathcal{M}}(x) = y\}] \tag{17}$$

We aim to show that the reward model $\boldsymbol{R}$ trained using $\mathcal{D}_i^c$ (positive samples) and $\hat{\mathcal{D}}_i^c$ (negative samples) will lead to a higher accuracy in the positive samples. The reward model objective is:

$$\max_{\boldsymbol{R}} \quad \mathbb{E}_{x \sim \mathcal{D}_i^c}[\boldsymbol{R}(x)] - \mathbb{E}_{x \sim \hat{\mathcal{D}}_i^c}[\boldsymbol{R}(x)] \tag{18}$$

By optimizing this objective, the reward model learns to assign higher rewards to samples from $\mathcal{D}_i^c$ and lower rewards to samples from $\hat{\mathcal{D}}_i^c$. This training process effectively filters out noise and emphasizes samples where the weak and strong models agree, leading to a classifier with higher accuracy.

Furthermore, because the consistent samples $\mathcal{D}_i^c$ are derived from both weak and strong datasets, they are more representative of high-confidence data, ensuring that the positive samples (i.e., samples with high $\boldsymbol{R}(x)$) are less noisy and have a distribution that is different from the noisy parts of the strong dataset ($\hat{\mathcal{D}}_i^c$).

Thus, we have shown that the classifier trained in this manner will produce positive samples with higher accuracy and a distribution that is further from the noisy strong datasets.

## D   MODEL PERFORMANCE ON DIFFERENT CAPABILITY

In this section, we present the performance of models of various scales across multiple capabilities. The results are shown in the following figures. The Figure (Figure 15) shows the performance of different-scale models on multi-capabilities tasks. Moreover, Figure 16 shows the performance of weak to strong generalization with single-capability and multi-capabilities.

## E   IMPACT OF DIFFERENT CAPABILITY

Figure 17 shows the performance of a strong model when trained with various weak supervision. In Figure 18 and Figure 19, we observe the results when the model's capabilities are deliberately weakened on the some dataset. Figure 20, Figure 21, Figure 22, and Figure 23 show the generalization performance of models when certain capabilities are removed.

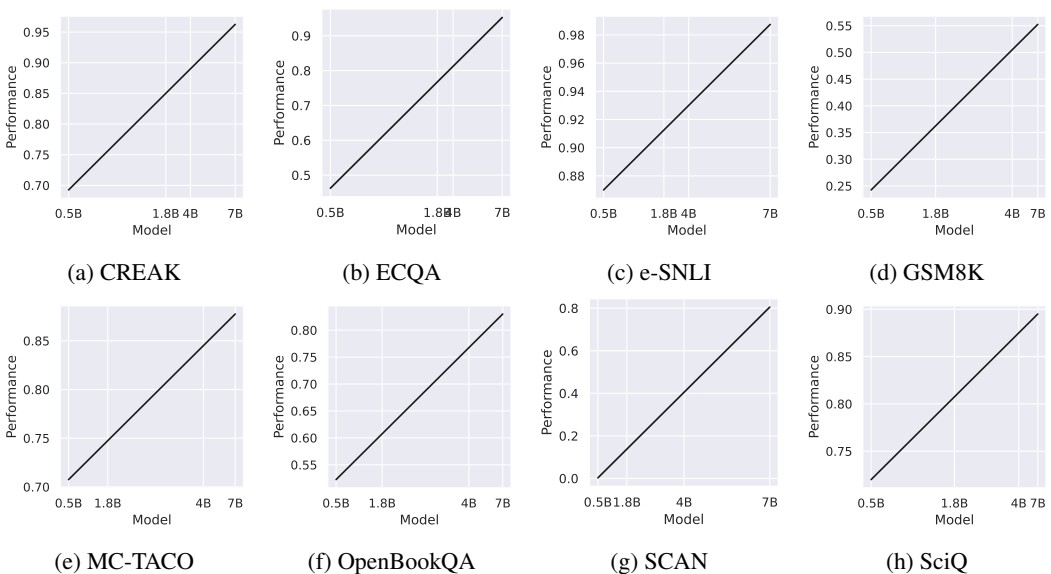

Figure 15: Performance of different-scale models on multi-capabilities tasks.

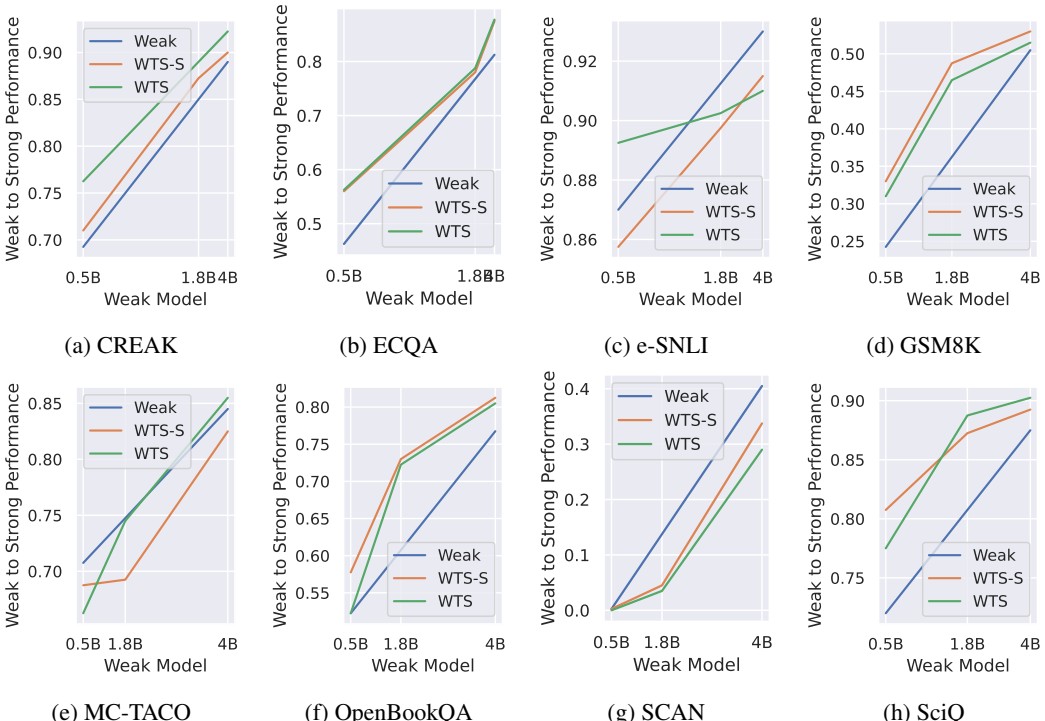

Figure 16: Performance of weak to strong generalization with single-capability and multi-capabilities.

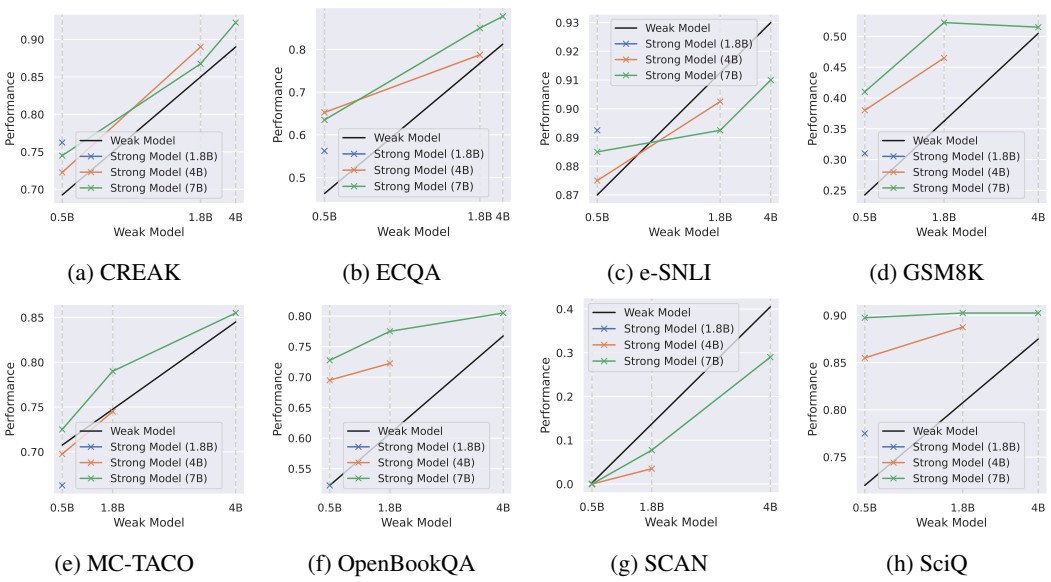

Figure 17: Performance of strong model trained on different weak supervision.

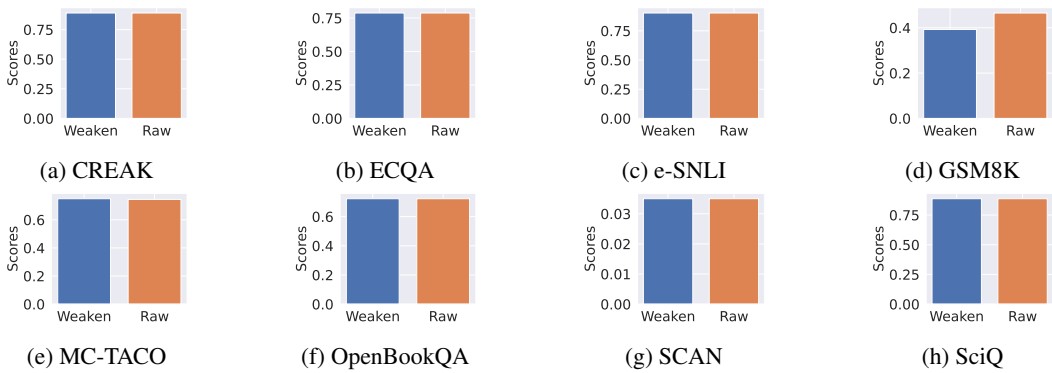

Figure 18: Results under weaken capability on GSM8k.

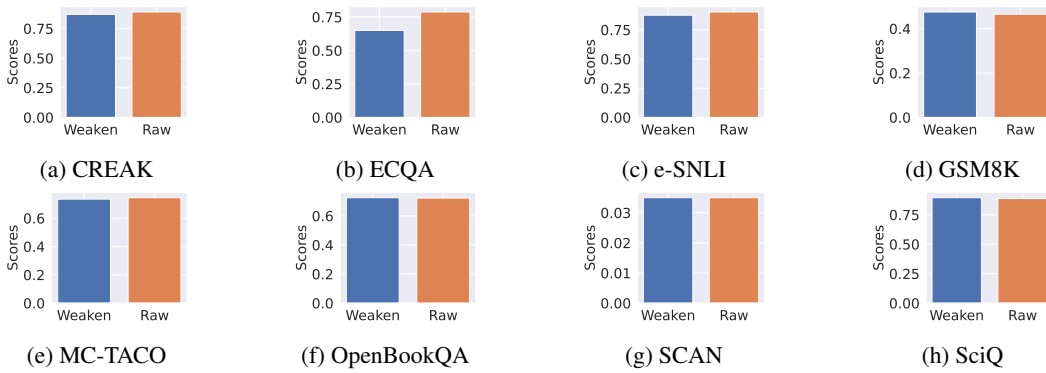

Figure 19: Results under weaken capabilities on ECQA and e-SNLI.

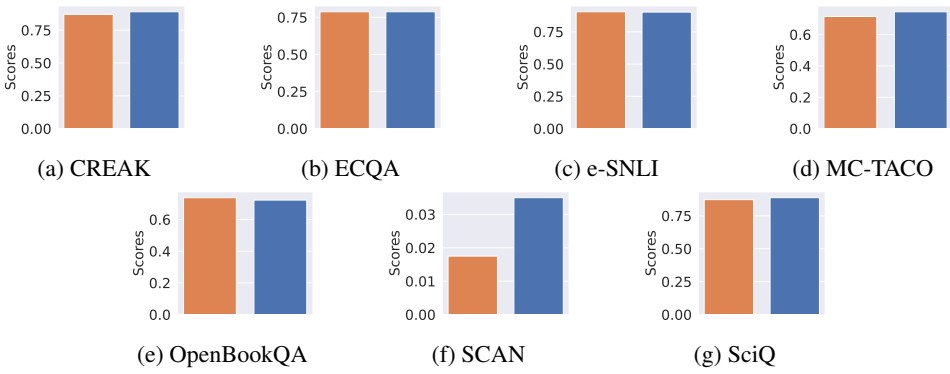

Figure 20: Results of 1.8B to 4B generalization under remove capability on GSM8k.

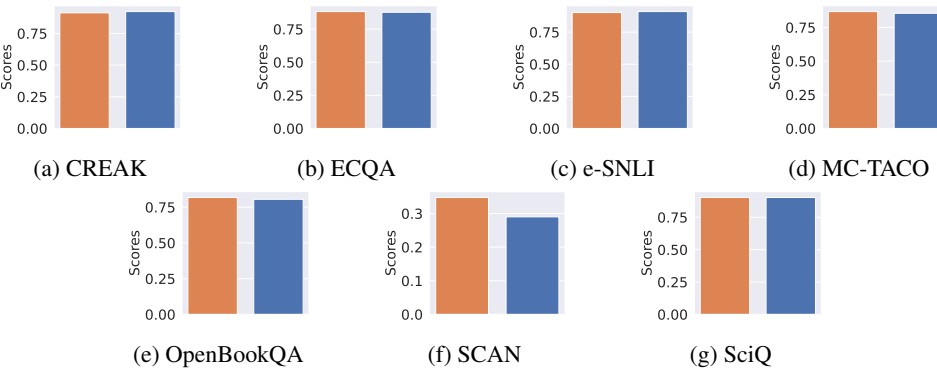

Figure 21: Results of 4B to 7B generalization under remove capability on GSM8k.

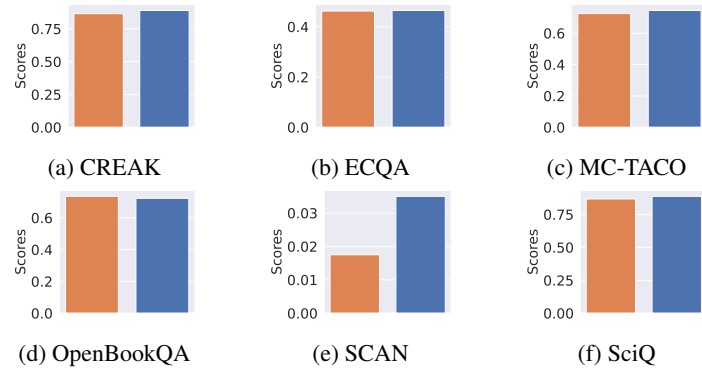

Figure 22: Results of 1.8B to 4B generalization under remove capabilities on ECQA and e-SNLI.

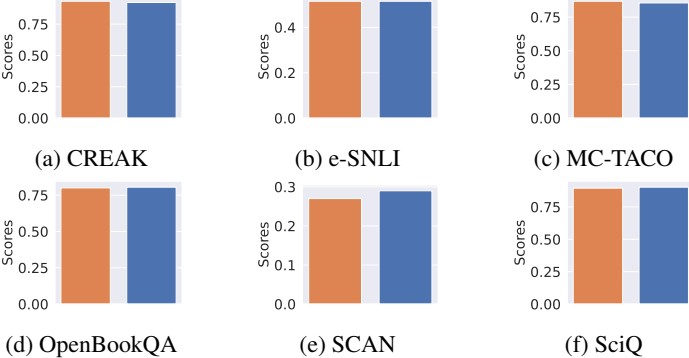

Figure 23: Results of 4B to 7B generalization under remove capabilities on ECQA and e-SNLI.

# F    CONSISTENCY ANALYSIS ON WEAK TO STRONG GENERALIZATION

This section presents an analysis of the consistency in prediction results between models of varying sizes. The analysis focuses on comparing the predictions of weaker models with those of stronger models. The following Figure 24, Figure 25, and Figure 26 illustrate the prediction results for different pairs of models.

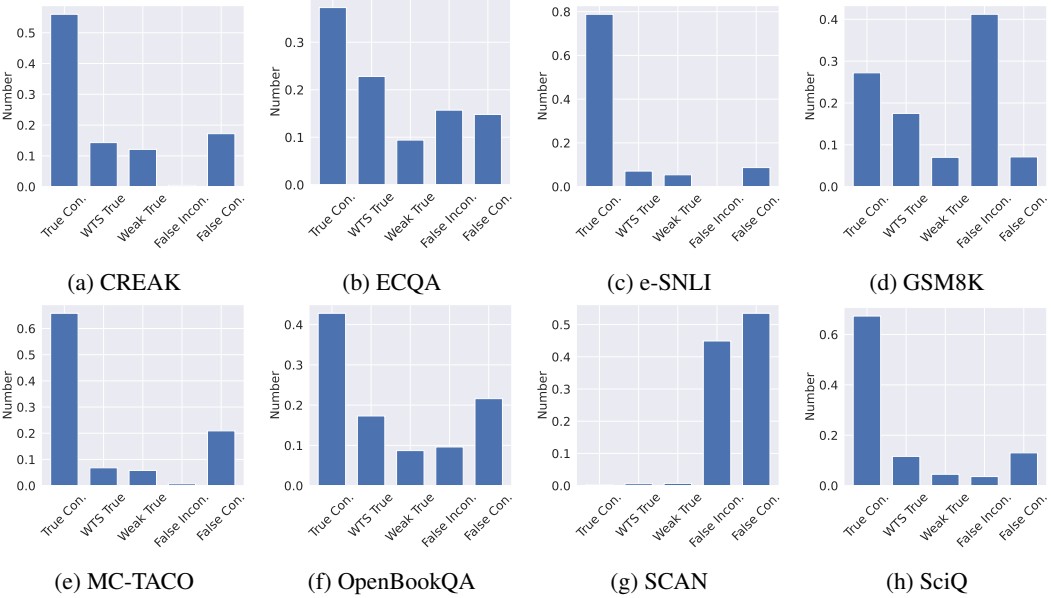

Figure 24: Prediction results of 1.8B strong model (WTS) and 0.5B weak model (Weak).

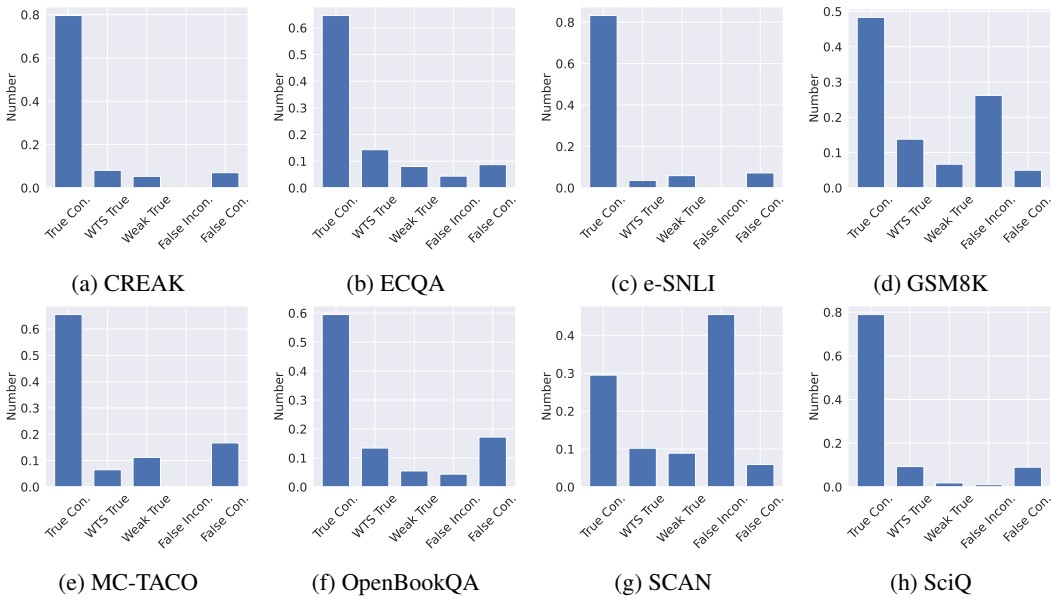

Figure 25: Prediction results of 4B strong model (WTS) and 1.8B weak model (Weak).

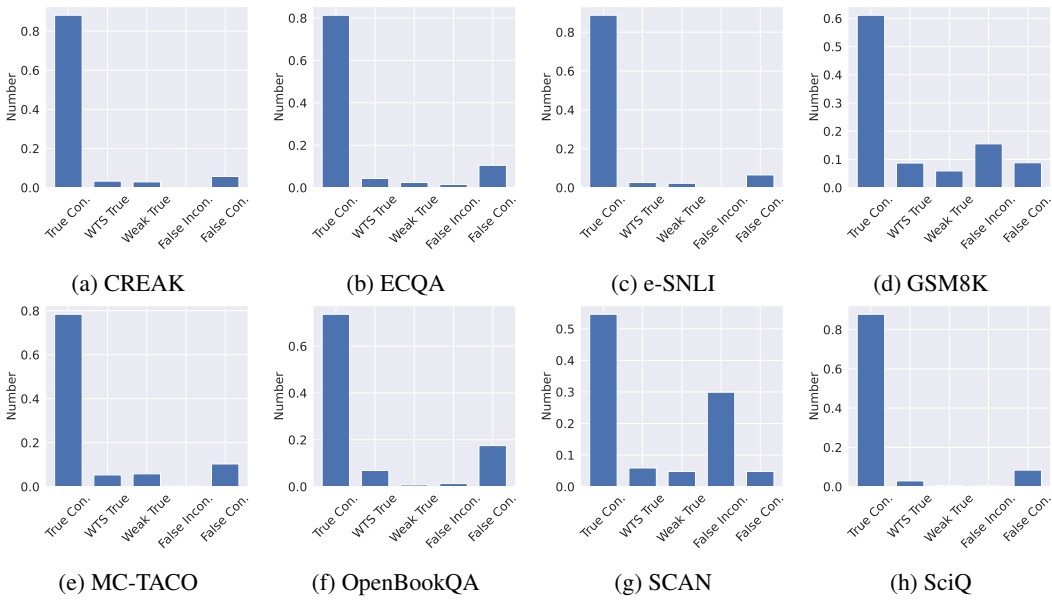

Figure 26: Prediction results of 7B strong model (WTS) and 4B weak model (Weak).

# G CONSISTENCY ANALYSIS ON STRONG MODEL SELF-BOOTSTRAPPING

This section presents a detailed consistency analysis of the self-bootstrapping used in training strong models. The bootstrapping involves using predictions from a previously trained strong model (WTS) to supervise the training of a new strong model (WTS-2). We evaluate the prediction consistency between these models across different scales in Figure 27, Figure 28, and Figure 29.

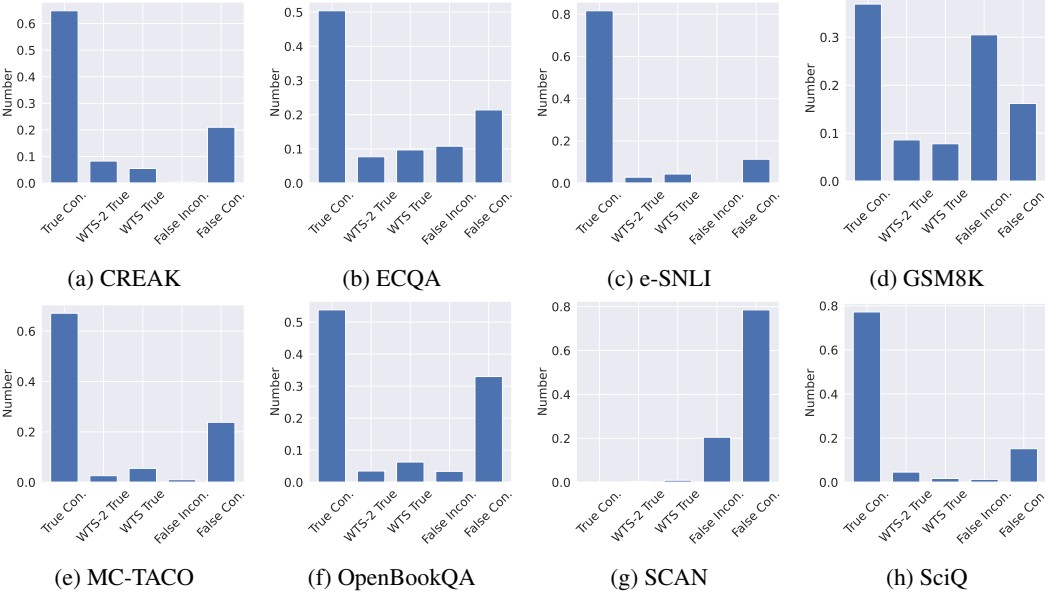

Figure 27: Prediction results of 1.8B strong model (WTS-2) and its supervision from strong model (WTS).

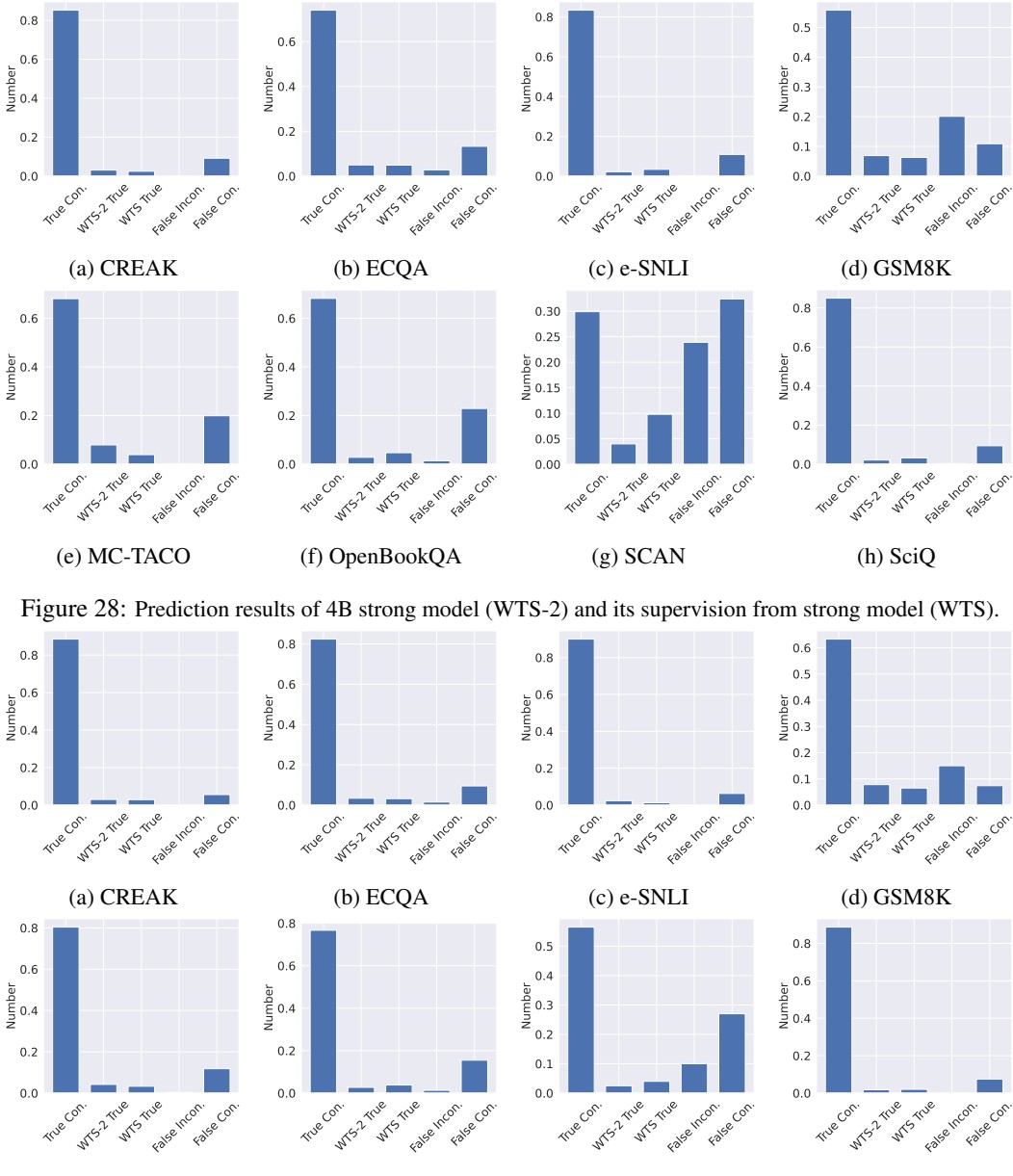

Figure 28: Prediction results of 4B strong model (WTS-2) and its supervision from strong model (WTS).

Figure 29: Prediction results of 7B strong model (WTS-2) and its supervision from strong model (WTS).

# H IMPACT OF WEAK DATA

This section examines the impact of weak data on model performance. The analysis is presented through Figure 30 and Figure 31 which illustrate the effects of training a strong model with clean and noisy weak datasets.

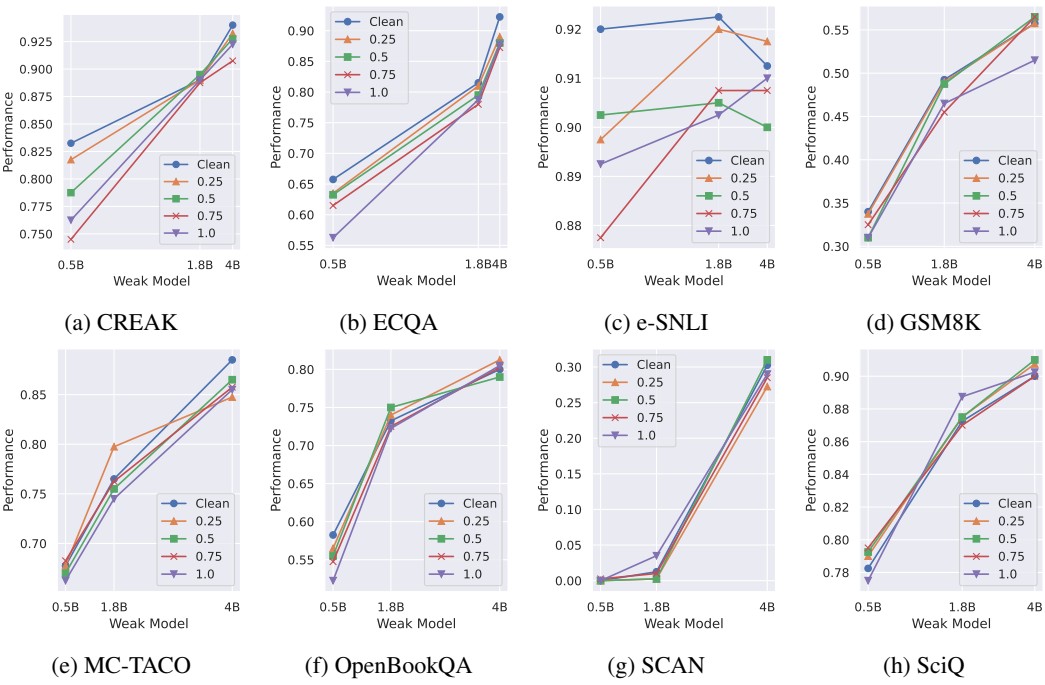

Figure 30: Each capability performance of strong model trained on clean samples and different proportions of noise samples of weak datasets.

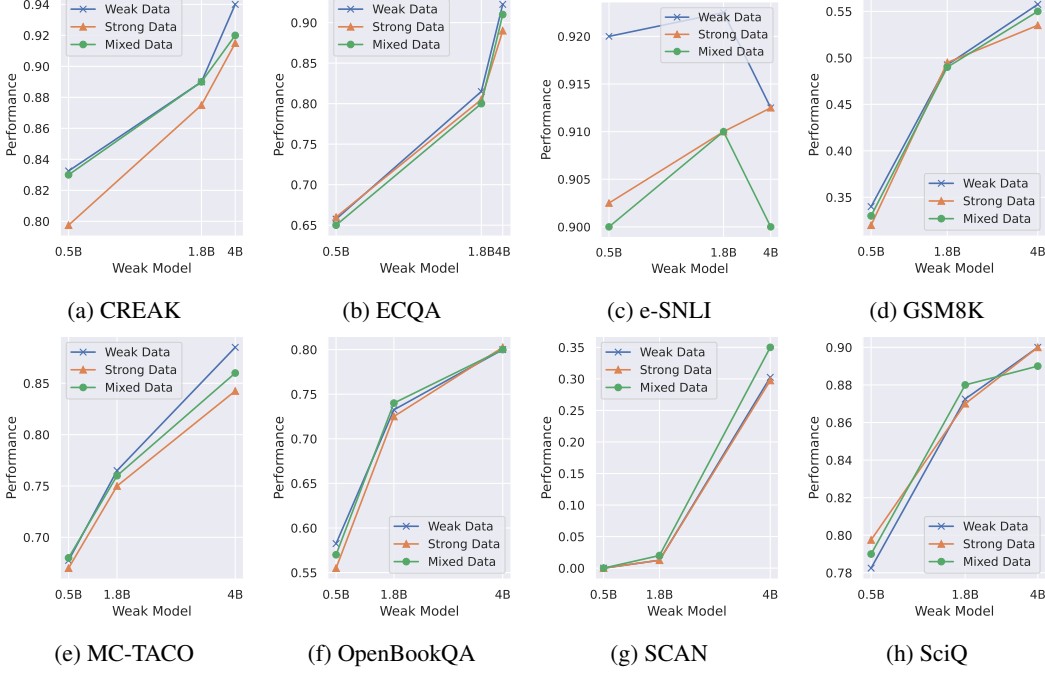

Figure 31: Each capability performance of strong model trained on weak datasets, strong datasets and their combination.

# I  PERFORMANCE ON EACH CAPABILITY

This section provides a detailed analysis of the weak to strong generalization performance of each capability. The results are illustrated in the Figure 32 and Figure 33.

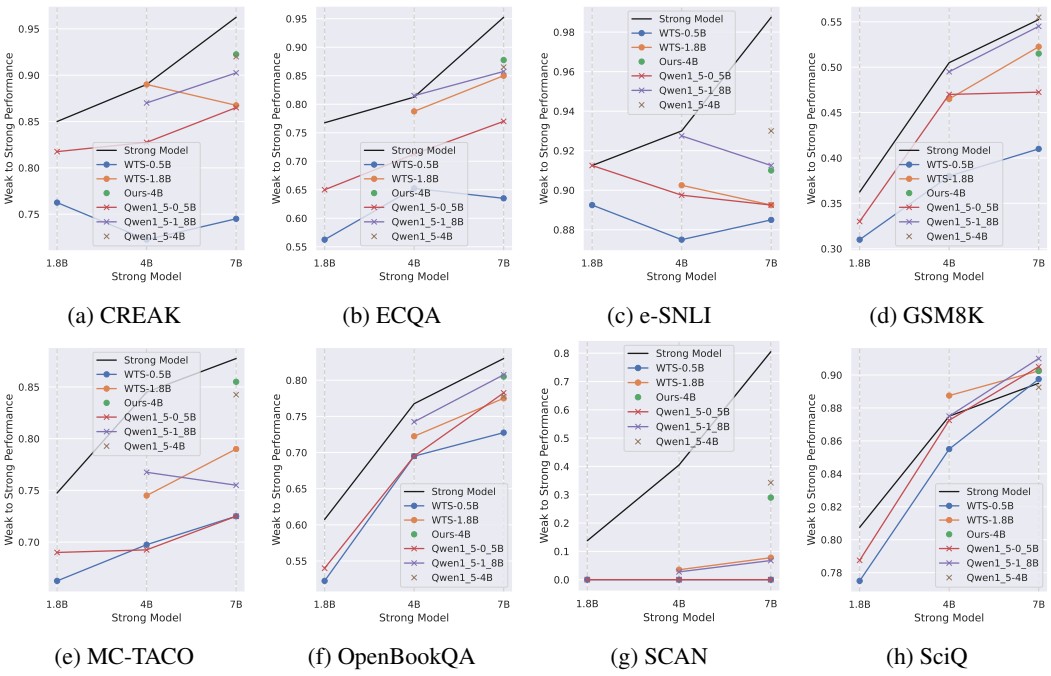

Figure 32: Each capability performance of weak to strong generalization.

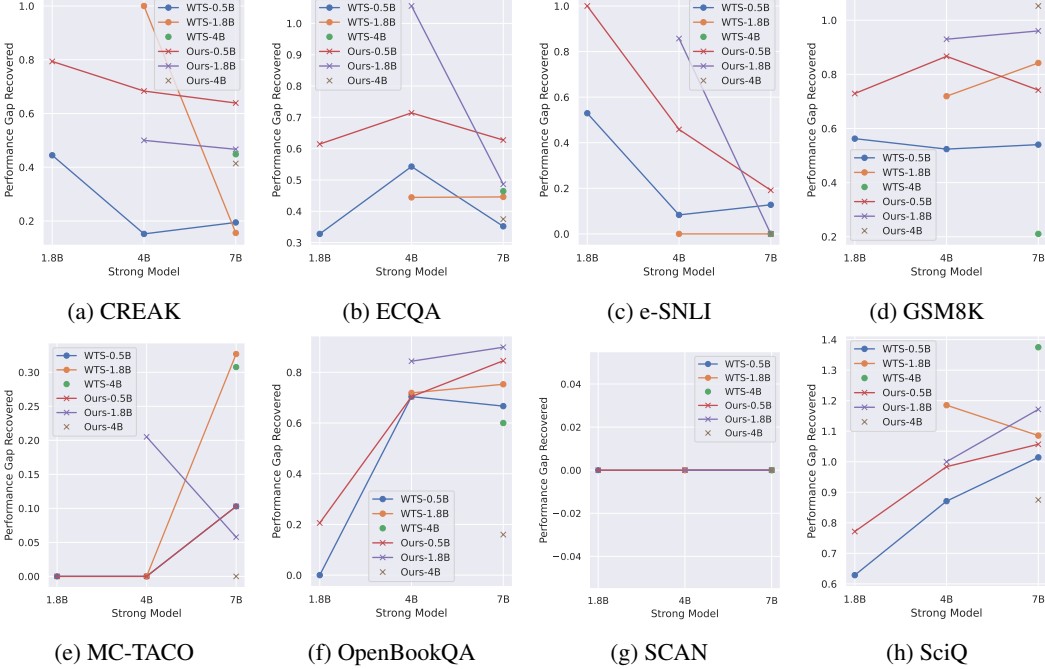

Figure 33: Performance gap recovered of weak to strong generalization on each capability.

## J  PERFORMANCE ON REWARD MODEL

In this section, we present the performance analysis of the reward model compared to weak and strong models across various capabilities. The analysis is visualized through Figure 34, Figure 35, Figure 36, and Figure 37, illustrating the accuracy and consistency of the models.

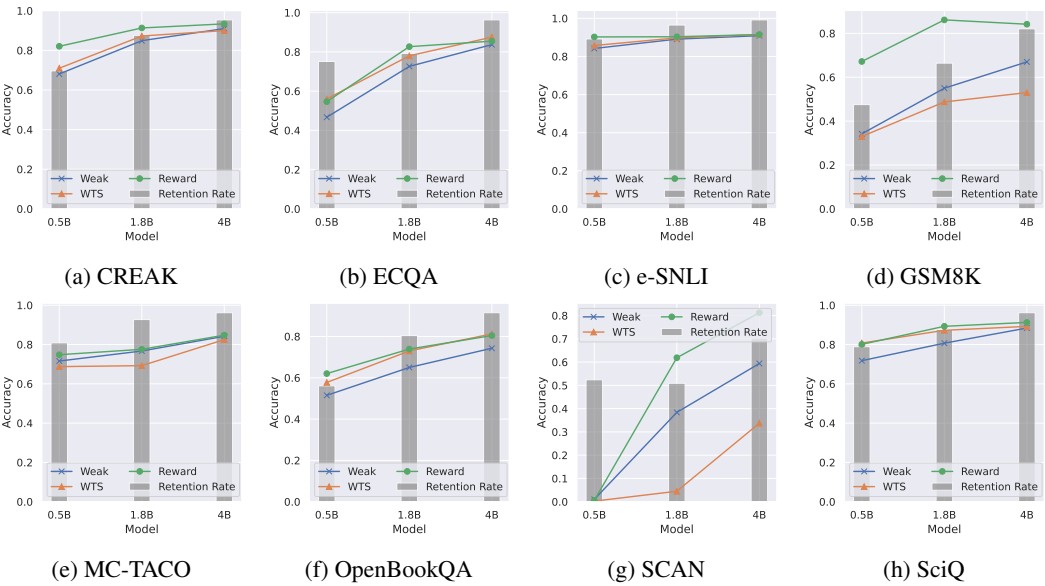

Figure 34: Accuracy of datasets from reward model, weak model and strong model on each capability.

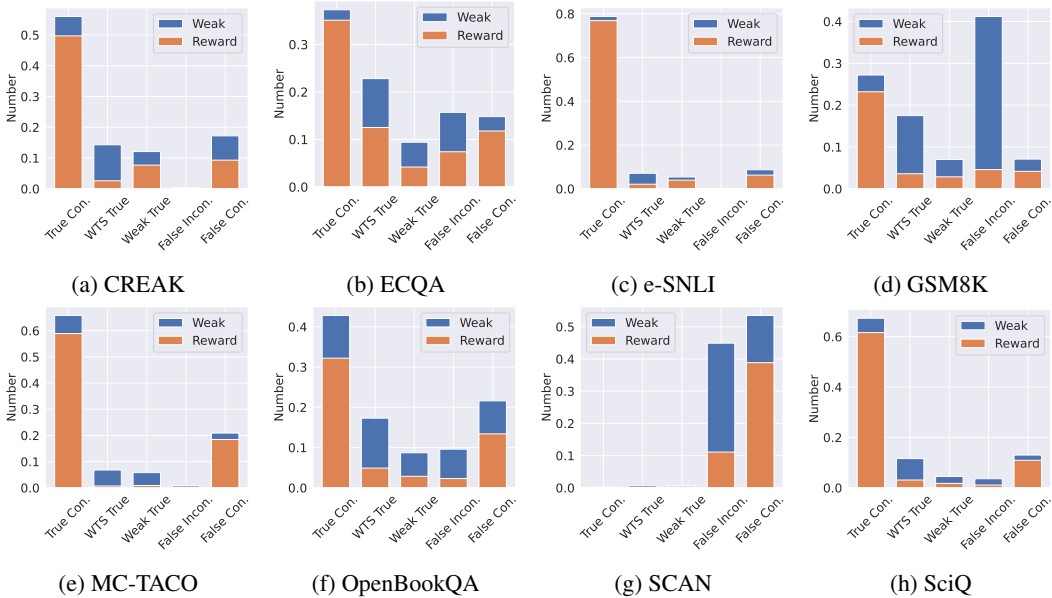

Figure 35: Performance of reward model (Reward) on (in)consistent part between 0.5B weak model (Weak) and 1.8B strong model.

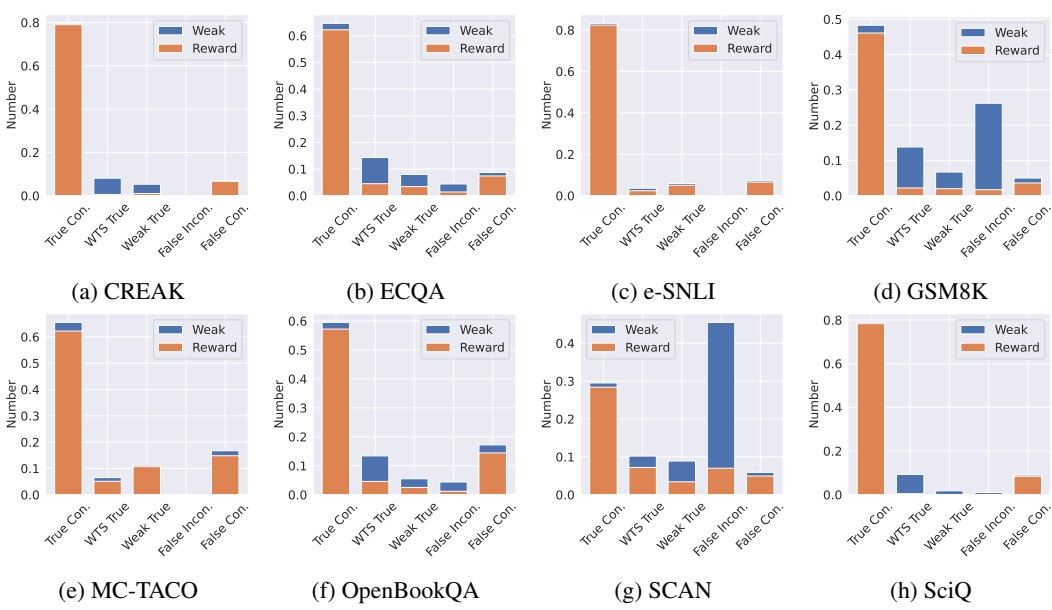

Figure 36: Performance of reward model (Reward) on (in)consistent part between 1.8B weak model (Weak) and 4B strong model.

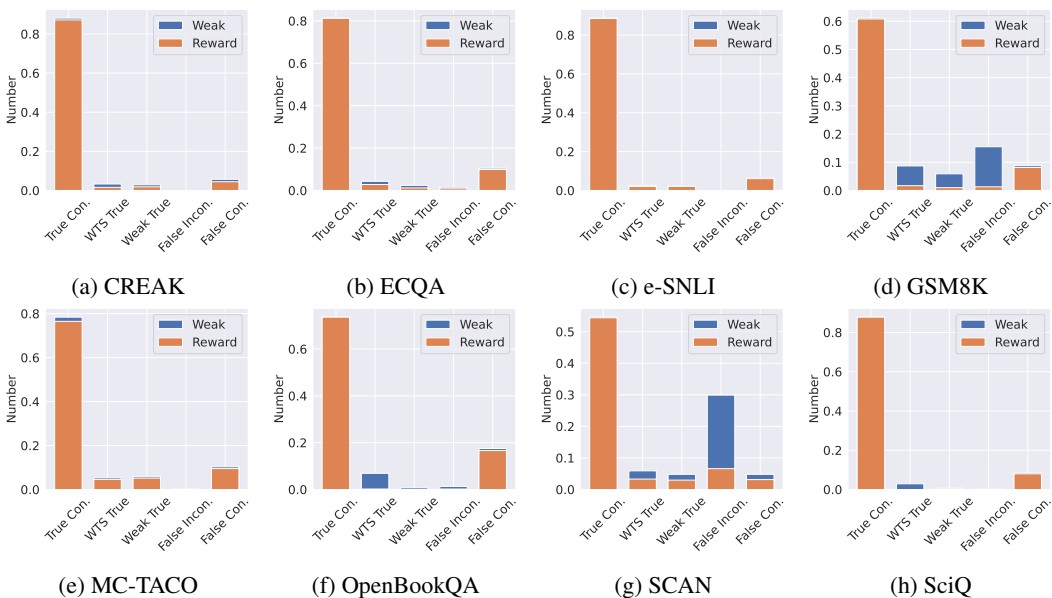

Figure 37: Performance of reward model (Reward) on (in)consistent part between 4B weak model (Weak) and 7B strong model.

## K    MORE DETAILED FIGURES

We have expanded on the content of Figure 1, Figure 7(Left), and Figure 7(Right) in greater detail. The detailed breakdown is presented in Figure 38, Figure 39, and Figure 40 below for better observation.

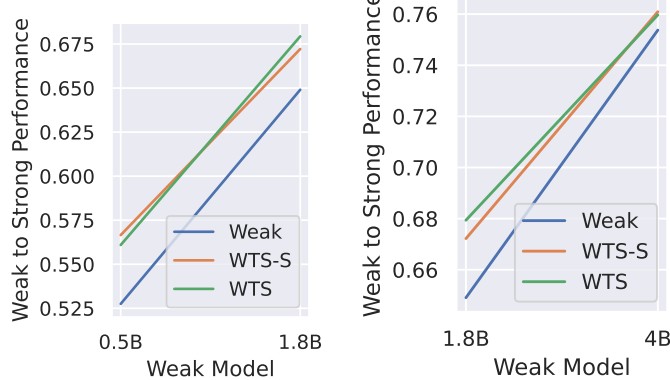

Figure 38: Performance of strong model with single-capability and multi-capabilities weak to strong generalization.

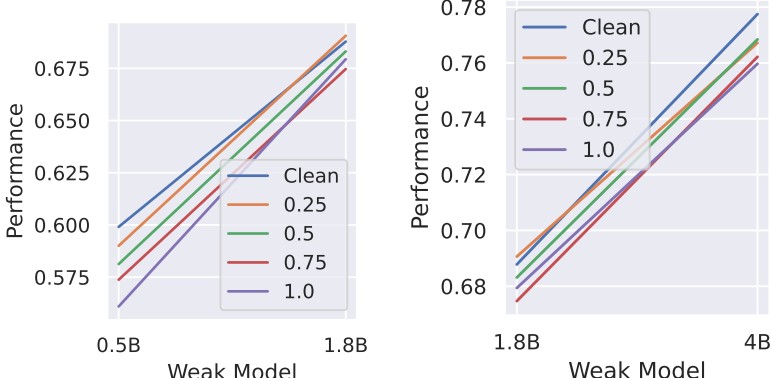

Figure 39: Performance of strong model trained on clean samples and different proportions of noise samples of weak datasets.

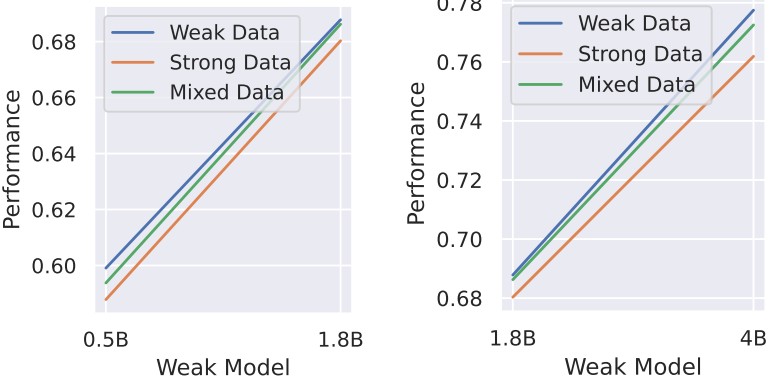

Figure 40: Performance of strong model trained on weak, strong, and combination datasets.

## L   ABLATION STUDY ON DATA SELECTION METHODS

We experiment to compare the effectiveness of our reward model-based data selection with a random selection baseline. Random selection was performed by choosing the same amount of data as selected by the reward model, as it is considered a strong baseline in LLM data selection. The results of these experiments are presented in Table 2.

| Weak Model | Strong Model | Random | Ours |
|---|---|---|---|
| 0.5B | 1.8B | 52.49 | 59.17 |
| 1.8B | 4B | 63.52 | 69.05 |
| 4B | 7B | 74.31 | 77.07 |

Table 2: Performance comparison between random data selection and reward model-based data selection.

The results show that random selection leads to a significant drop in performance compared to our reward model-based approach. This performance gap can be attributed to the reward model's ability to identify and prioritize higher-quality training samples, which is critical for achieving better generalization. These findings underscore the importance of the reward model in improving the data selection process.

## M   PERFORMANCE VERIFICATION WITH IN-CONTEXT LEARNING

To better verify the performance of the strong model before training, we assessed its capabilities using in-context learning (ICL). Specifically, we evaluated the strong model's performance before training by providing 3 ground truth examples for ICL. The experimental results are presented in Table 3.

| Weak Model | Strong Model | Ours | WTS | ICL |
|---|---|---|---|---|
| 0.5B | 1.8B | 59.19 | 56.10 | 45.92 |
| 1.8B | 4B | 69.07 | 68.00 | 58.38 |
| 4B | 7B | 77.06 | 76.11 | 70.02 |

Table 3: Performance comparison between our method, standard weak-to-strong generalization (WTS), and in-context learning (ICL).

The results demonstrate that weak-to-strong generalization (including both our method and the standard WTS approach) significantly outperforms ICL across all model sizes. This performance gap highlights the limitations of ICL, which relies solely on a few examples and cannot fully learn the task's intent. In contrast, weak-to-strong generalization effectively learns the task's intent from noisy weak data and exhibits remarkable resistance to noise. This robustness underscores the superiority of weak-to-strong generalization in scenarios where data quality is inconsistent or noisy, further validating its effectiveness for large-scale model training.

