# OpenReview forum: "Weak to Strong Generalization for Large Language Models with Multi-capabilities"
_ICLR.cc/2025/Conference — ICLR 2025 Poster_

### Official Review · Reviewer_orGJ · 2024-11-04

**Soundness:** 3
**Presentation:** 4
**Contribution:** 4
**Rating:** 6
**Confidence:** 4

**Summary:**

This paper investigates the weak-to-strong generalization capabilities of LLMs across multiple capabilities. Through extensive experiments, key findings include the relative independence of capabilities, the significant impact of data quality on weak supervision, and the overconfidence of strong models in specific knowledge areas, leading to performance degradation. To enhance weak-to-strong generalization, the authors propose a novel training framework. This framework uses reward models to select valuable weak supervision data, generated by weaker models, for training stronger models. A reward model is trained to identify datasets that differ in distribution from the strong datasets, ensuring diversity. A two-stage training method is then applied, allowing the strong model to learn from both weak and selected datasets, resulting in improved multi-capabilities weak-to-strong generalization.

**Strengths:**

The authors present how weak supervision affects strong models, revealing that strong models do not always follow the guidance of weak models. This insight is crucial for understanding the limitations and potential pitfalls of using weak supervision. And the proposed training framework, which uses reward models to select valuable data for weak supervision, is innovative. It addresses the issue of overconfidence in strong models and leverages the diversity of weak datasets to improve generalization. This paper clearly outlines the steps involved in the proposed framework, making it easy to understand.

**Weaknesses:**

Although the paper points out that overconfidence is an important problem in strong models, it does not provide a detailed description of the reward model used to select valuable weak supervision data, such as the reward model's framework and training data. In addition, this paper uses 'performance' to refer to the experimental results of the comparison models in several places, without clearly describing the specific evaluation metrics of the performance. Clearer metrics would help readers better understand the improvements achieved by the proposed framework.

**Questions:**

The paper explores the weak to strong generalization for LLMs across multiple capabilities. However, there are several aspects that require further clarification and improvement to strengthen the validity and robustness of the proposed method. Below are my detailed comments:
1. The paper frequently uses the term "performance" to describe the experimental results of the comparison models without clearly specifying the particular evaluation metrics. Could the authors clarify what is meant by "performance"? Is it accuracy, or another metric? Please provide a clear definition and the specific metrics used.
2. In Equation 6, why is Dci considered low-confidence data? Is it because this data is generated by a strong model through relabeling? Could the rationale behind treating it as low-confidence data be to enhance data diversity and thereby prevent model collapse?
3. Could you provide more details about the reward model R(x)? Specifically, what are the training data and model architecture used for R(x)? Are the reward model, weak model, and strong model all instances of Qwen 1.5, or do they differ in their configurations?
4. In the experiments conducted with different model sizes, were the sizes of the datasets (e.g., the labeled dataset and the unlabeled dataset) varied? For instance, was a larger amount of data used for the 4B model compared to the 1.8B model?

---

> ### Author Response · Authors · 2024-11-25
> **Response to Reviewer orGJ**
>
> Thanks a lot for reviewing our manuscript and thanks for your positive comments (i.e., the paper's insight is crucial for understanding the limitations and potential pitfalls of using weak supervision). We are sorry for the confusion and unclear expression in the previous version (the main revisions are marked with blue text in the pdf). We have addressed each of the comments and suggestions below.
>
> > The paper frequently uses the term "performance" to describe the experimental results of the comparison models without clearly specifying the particular evaluation metrics. Could the authors clarify what is meant by "performance"? Is it accuracy, or another metric? Please provide a clear definition and the specific metrics used.
>
> Thank you for your suggestion. In "5.1 EXPERIMENTAL SETUPS", under "Experimental Details", we have further elaborated on the specific metrics used for performance evaluation. The performance refers to accuracy, where a correct prediction exactly matches the ground truth answer.
>
> > In Equation 6, why is Dci considered low-confidence data? Is it because this data is generated by a strong model through relabeling? Could the rationale behind treating it as low-confidence data be to enhance data diversity and thereby prevent model collapse?
>
> $\hat{D}_i^c$ is regarded as low-confidence data because, as observed in Figure 5, the predictions of the strong model that disagree with those of the weak model have lower accuracy. Additionally, as shown in the results from Figure 6, the data predicted by the strong model exhibits lower diversity. Therefore, treating $\hat{D}_i^c$ as negative samples also helps prevent model collapse.
>
> > Could you provide more details about the reward model R(x)? Specifically, what are the training data and model architecture used for R(x)? Are the reward model, weak model, and strong model all instances of Qwen 1.5, or do they differ in their configurations?
>
> Thank you for your suggestions! In "5.1 EXPERIMENTAL SETUPS" under "Experimental Details", we further elaborate that the reward models are initialized from the strong model, i.e., Qwen-1.5, and they maintain the same parameters. Additionally, in "4.2 TRAINING REWARD MODELS", we have already mentioned that we choose $D_i^c$ as positive samples and $\hat{D}_i^c$ as negative samples for reward model (classifier) training.
>
> > In the experiments conducted with different model sizes, were the sizes of the datasets (e.g., the labeled dataset and the unlabeled dataset) varied? For instance, was a larger amount of data used for the 4B model compared to the 1.8B model
>
> Our dataset includes both the labeled dataset and the unlabeled dataset, which are independent of model size. The same dataset is used across models of different sizes, following the setup in [1].
>
> [1] Weak-to-strong generalization: Eliciting strong capabilities with weak supervision

---

> ### Author Response · Authors · 2024-11-28
> **Official Comment by Authors**
>
> Dear Reviewer orGJ,
>
> We greatly appreciate the time and effort you have taken to review our work. Since the author-reviewer discussion is closing very soon, could you please check our response to see whether it addressed your concerns? We remain open to further discussion and revisions.
>
> Thanks,
>
> The authors

---

> > ### Author Response · Authors · 2024-12-04
> > **Official Comment by Authors**
> >
> > Dear Reviewer orGJ,
> >
> > We greatly appreciate the time and effort you have taken to review our work. We kindly hope that you can take some time to reconsider the rating of our paper based on our responses above. Thank you!
> >
> > Best regards,
> >
> > The authors

---

### Official Review · Reviewer_fdJr · 2024-11-04

**Soundness:** 3
**Presentation:** 3
**Contribution:** 3
**Rating:** 6
**Confidence:** 4

**Summary:**

This paper conducts a thorough exploratory study on multi-capability alignment within the realm of weak supervision for strong model training. The authors make several key observations, such as the relative independence of different tasks when weak supervision is applied and the tendency of models to become overly reliant on limited supervision through bootstrapping. These insights highlight the challenges and limitations of current approaches. Based on these findings, the authors propose a theoretically and experimentally grounded two-stage training framework, which includes training reward models for data selection and training strong models. This framework effectively enables learning from weak datasets and the selected data, addressing the inconsistencies between weak and strong datasets.

**Strengths:**

1. The research topic is novel and meaningful, providing valuable insights into multi-capability alignment under the concept of super-alignment.
2. The writing is rigorous and standardized, with a deep and thorough analysis that enhances the overall clarity of the paper.
3. The proposed two-stage approach effectively addresses key challenges within the research paradigm, supported by solid theoretical analysis and experimental validation.

**Weaknesses:**

1. While the paper exhibits excellent logical flow and robust experimentation, the presentation of key figures, such as Figure 1, lacks clarity and attention to detail. The meanings of the axes, calculation formulas, and the implications of the legends are somewhat ambiguous, leading to confusion regarding the data sources. The overlapping results of different methods detract from the visual appeal, and the depiction of performance proportional to model size raises questions, especially when the same-sized methods are represented in different colors. It is advisable to optimize this section for better clarity and presentation.

2. The width of the model size representation could be expanded to better align with the theme of super-alignment, thereby more effectively demonstrating the robustness of the proposed method.

**Questions:**

As mentioned in the above weaknesses.

---

> ### Author Response · Authors · 2024-11-25
> **Response to Reviewer fdJr**
>
> Thank you very much for taking the valuable time to review our manuscript and thanks for your positive comments (i.e., providing valuable insights into multi-capability alignment under the concept of super-alignment, and the proposed approach effectively addresses key challenges, supported by solid theoretical analysis and experimental validation). We are sorry for the confusion and unclear expression in the previous version (the main revisions are marked with blue text in the pdf). We have addressed each of the comments and suggestions. Please refer to our responses below for details.
>
> > While the paper exhibits excellent logical flow and robust experimentation, the presentation of key figures, such as Figure 1, lacks clarity and attention to detail. The meanings of the axes, calculation formulas, and the implications of the legends are somewhat ambiguous, leading to confusion regarding the data sources. The overlapping results of different methods detract from the visual appeal, and the depiction of performance proportional to model size raises questions, especially when the same-sized methods are represented in different colors. It is advisable to optimize this section for better clarity and presentation. The width of the model size representation could be expanded to better align with the theme of super-alignment, thereby more effectively demonstrating the robustness of the proposed method.
>
> Thank you for your constructive suggestion. It is greatly helpful for improving the paper's presentation. We have added more detailed figures in the appendix to enhance clarity and address the issues raised.

---

> > ### Author Response · Authors · 2024-11-28
> > **Official Comment by Authors**
> >
> > Dear Reviewer fdJr,
> >
> > We greatly appreciate the time and effort you have taken to review our work. Since the author-reviewer discussion is closing very soon, could you please check our response to see whether it addressed your concerns? We remain open to further discussion and revisions.
> >
> > Thanks,
> >
> > The authors

---

> > > ### Author Response · Authors · 2024-12-04
> > > **Official Comment by Authors**
> > >
> > > Dear Reviewer fdJr,
> > >
> > > We greatly appreciate the time and effort you have taken to review our work. We kindly hope that you can take some time to reconsider the rating of our paper based on our responses above. Thank you!
> > >
> > > Best regards,
> > >
> > > The authors

---

### Official Review · Reviewer_5nVp · 2024-11-04

**Soundness:** 2
**Presentation:** 3
**Contribution:** 3
**Rating:** 5
**Confidence:** 4

**Summary:**

This paper first investigates multi-capability weak-to-strong generalization in large language models (LLMs). It then confirms that diverse, high-quality data from weak models improves generalization for strong models. To enhance this process, the authors introduce a two-stage training framework that uses reward models to select optimal weak supervision data.

**Strengths:**

1. As the first study to investigate multi-capability weak-to-strong generalization, it can provide some relevant conclusions for this field.
2. This study explores data selection directions for weak-to-strong generalization and proposes corresponding methods to improve performance.

**Weaknesses:**

1. The research on multi-capability weak-to-strong generalization seems rather isolated from the later stages of weak data selection and also lacks a connection to the proposed method. It is a relatively shallow and separate exploration.
2. The validation of self-bootstrapping's ineffectiveness is insufficient. Concerns are detailed in the 'Questions' section.
3. The necessity and effectiveness of the reward model lack sufficient validation and investigation. For example, an ablation study on Equation 7 was not conducted. Additional concerns are detailed in the 'Questions' section.
4. The necessity of the warm-up stage lacks further validation. For example, running only 2 epochs might be insufficient, as the model's optimal performance is achieved after a total of 4 epochs, suggesting that it may be the total training time, rather than the distinct nature of the two stages, that drives performance improvements.

**Questions:**

1. I am curious whether the second version of the strong model in Section 3.5 is further trained on the existing trained strong model, or if it is untrained. If an untrained strong model is used to relabel data and then continues to train itself, would that improve performance? Additionally, while greedy search can exacerbate the issue of overconfidence, would sampling help mitigate this? Specifically, by using multiple sampling and voting (self-consistency) to select more diverse and accurate samples, could it lead to greater self-bootstrapping gains?
2. Is a separate reward model trained for each weak-to-strong training instance? If so, wouldn’t this make the overall training pipeline somewhat lengthy? Additionally, I understand that the reward model’s training data and inference data are identical, so how many weak data points with the same labels are actually filtered out? If not, how does the reward model generalize?
3. To what extent might rule-based data selection methods serve as alternatives to the reward model? For instance, could calculating semantic similarity or n-gram redundancy be effective approaches?
4. What is the performance of the strong model before training? How does it compare with the supervised trained weak model and the strong model after weak-to-strong training? I am curious about the significance of studying weak-to-strong generalization.

---

> ### Author Response · Authors · 2024-11-25
> **Response to Reviewer 5nVp**
>
> Thank you very much for taking the valuable time to review our manuscript and thanks for your positive comments (i.e., as the first study to investigate multi-capability weak-to-strong generalization, it can provide some relevant conclusions for this field). We are sorry for the confusion and unclear expression in the previous version (the main revisions are marked with blue text in the pdf). We have addressed each of the comments and suggestions. Please refer to our responses below for details.
>
> > The research on multi-capability weak-to-strong generalization seems rather isolated from the later stages of weak data selection and also lacks a connection to the proposed method. It is a relatively shallow and separate exploration.
>
> We observed from the research on multi-capability weak-to-strong generalization that using data generated by a strong model for its further training can lead to performance degradation, known as model collapse. Additionally, the overlapping parts between the weak dataset and the strong dataset exhibit higher accuracy. Based on these insights, we propose a data selection method.
>
> > The necessity of the warm-up stage lacks further validation. For example, running only 2 epochs might be insufficient, as the model's optimal performance is achieved after a total of 4 epochs, suggesting that it may be the total training time, rather than the distinct nature of the two stages, that drives performance improvements.
>
> Our experiments consistently maintain a training regime of 2 epochs, with each of the two stages lasting for one epoch.
>
> > I am curious whether the second version of the strong model in Section 3.5 is further trained on the existing trained strong model, or if it is untrained. If an untrained strong model is used to relabel data and then continues to train itself, would that improve performance? Additionally, while greedy search can exacerbate the issue of overconfidence, would sampling help mitigate this? Specifically, by using multiple sampling and voting (self-consistency) to select more diverse and accurate samples, could it lead to greater self-bootstrapping gains?
>
> The second version of the strong model in Section 3.5 is trained on the untrained strong model. Under in-context learning (ICL), the untrained strong model (7B) achieves only 70.02% accuracy, which is insufficient compared to the 76.11% accuracy of WTS (4B to 7B) to support relabeling. We attempted majority voting across three sampling runs, but the results remained consistent with greedy search and failed to mitigate the overconfidence issue.
>
> > Is a separate reward model trained for each weak-to-strong training instance? If so, wouldn’t this make the overall training pipeline somewhat lengthy? Additionally, I understand that the reward model’s training data and inference data are identical, so how many weak data points with the same labels are actually filtered out? If not, how does the reward model generalize?
>
> Training the reward model requires only about 1/5 of the computational resources needed for WTS training. Regarding the retention rate of the weak dataset, we have already presented it in Figure 11.
>
> > To what extent might rule-based data selection methods serve as alternatives to the reward model? For instance, could calculating semantic similarity or n-gram redundancy be effective approaches?
>
> Rule-based data selection methods cannot effectively identify correct samples. In contrast, our approach not only selects data that is more diverse relative to the strong model but also maintains higher accuracy, as shown in Figure 11.
>
> > What is the performance of the strong model before training? How does it compare with the supervised trained weak model and the strong model after weak-to-strong training? I am curious about the significance of studying weak-to-strong generalization.
>
> To better verify the performance of the strong model before training, we assessed the performance of the strong model before training with in-context learning (ICL). The experimental results are as follows:
>
> | Weak Model | Strong Model | Ours  | WTS   | ICL   |
> |------------|--------------|-------|-------|-------|
> | 0.5B       | 1.8B         | 59.19 | 56.10 | 45.92 |
> | 1.8B       | 4B           | 69.07 | 68.00 | 58.38 |
> | 4B         | 7B           | 77.06 | 76.11 | 70.02 |
>
> In the experiments, 3 ground truth examples were used for ICL. The results clearly show that weak-to-strong generalization (including both our method and the standard weak-to-strong (WTS) approach) significantly outperforms ICL. Unlike ICL, as shown in Figure 5, weak-to-strong generalization effectively learns the task's intent from noisy weak data, demonstrating remarkable resistance to noise.

---

> > ### Author Response · Authors · 2024-11-28
> > **Official Comment by Authors**
> >
> > Dear Reviewer 5nVp,
> >
> > We greatly appreciate the time and effort you have taken to review our work. Since the author-reviewer discussion is closing very soon, could you please check our response to see whether it addressed your concerns? We remain open to further discussion and revisions.
> >
> > Thanks,
> >
> > The authors

---

> ### Author Response · Authors · 2024-12-04
> **Official Comment by Authors**
>
> Dear Reviewer 5nVp,
>
> We greatly appreciate the time and effort you have taken to review our work. We kindly hope that you can take some time to reconsider the rating of our paper based on our responses above. Thank you!
>
> Best regards,
>
> The authors

---

### Official Review · Reviewer_r2a7 · 2024-11-04

**Soundness:** 2
**Presentation:** 2
**Contribution:** 2
**Rating:** 5
**Confidence:** 3

**Summary:**

This paper analyzes a multi-capabilities setting under the weak to strong generalization paradigm. Based on this analysis, a training framework for this paradigm is proposed, which includes a reward model for selecting valuable weak supervision data, followed by a two-stage training of a strong model on both the weak dataset and the selected dataset. Further discussion on the reward model is also provided.

**Strengths:**

The paper analyzes the current gaps in research on the weak to strong generalization paradigm from a "super-alignment" perspective, proposing a new setup for experimental analysis and providing a novel training approach under this setup.

**Weaknesses:**

1. The significance of the multi-capabilities setting is unclear. In Section 3.3, the paper concludes that since the quality of weak data is relatively independent across different capabilities and does not interfere with each other during generalization, the setup seems only quantitatively different from a single capability setup. Therefore, is it necessary to study this setup separately? It is recommended to clarify the differences between the single capability setup and the one proposed in this paper.

2. The paper does not clearly define the diversity of the weak dataset. It is mentioned that the diversity of a weak dataset can lead to better results as previous works. The experiment the third chapter focuses on the consistency of results generated by weak and strong models, which is typically not an explicit measure of diversity.

3. The paper lacks ablation experiments. It only presents the performance without using the reward model for data selection. It is suggested to conduct experiments by randomly selecting the same amount of data as picked by the reward model (as random selection is a strong baseline in LLM data selection).

4. The legends in the figures are unclear, and some contain labeling errors. In the first three sections of Chapter 3, "weak model" is used as the axis label, yet most of the time, the data shown is based on the performance of the strong model, causing confusion. Additionally, there are errors in the numbering of Figures 3 and 4 in Section 3.3. Besides, there is inconsistent usage of "weak to strong generalization" and "weak-to-strong generalization."

**Questions:**

1. What is the connection between the conclusion regarding the independence of different capabilities mentioned in Weakness 1 and the subsequent methodology and experiments?

2. In the scaling experiments, it is observed that as model size increases, the difference between the proposed method and the baseline becomes smaller. Considering the reward model's size is consistent with that of the strong model and the data used for the reward model is from the weak and strong model training, does the proposed method have certain limitations? (In simple terms, is the reward model scalable?)

---

> ### Author Response · Authors · 2024-11-25
> **Response to Reviewer r2a7**
>
> Thanks a lot for reviewing our manuscript and thanks for your positive comments (i.e., analyzing the current gaps in research on the weak to strong generalization, proposing a new setup for experimental analysis, and providing a novel training approach under this setup). We are sorry for the confusion and unclear expression in the previous version (the main revisions are marked with blue text in the pdf). We have addressed each of the comments and suggestions below.
>
> > The significance of the multi-capabilities setting is unclear. In Section 3.3, the paper concludes that since the quality of weak data is relatively independent across different capabilities and does not interfere with each other during generalization, the setup seems only quantitatively different from a single capability setup. Therefore, is it necessary to study this setup separately? It is recommended to clarify the differences between the single capability setup and the one proposed in this paper.
>
> Our study primarily explores whether weak-to-strong generalization can be achieved across multiple capabilities and how these capabilities are learned in LLMs. Our experiments confirm that capabilities are relatively independent from one another. Research on multi-capabilities setup is crucial because weak-to-strong generalization can be effectively applied to large-scale models with substantial parameter counts. Compared to single-capability strong models, greater emphasis should be placed on multi-capability large models like GPT-4 and Gemini, which are more practical for real-world scenarios.
>
> > The paper does not clearly define the diversity of the weak dataset. It is mentioned that the diversity of a weak dataset can lead to better results as previous works. The experiment the third chapter focuses on the consistency of results generated by weak and strong models, which is typically not an explicit measure of diversity.
>
> From Figures 5 and 6, we observe that fine-tuning untrained strong model on datasets generated by the strong model leads to more consistent inference results. This indicates that the self-bootstrapping learning of a strong model tends to reduce the diversity of its inference outcomes.
>
> > The paper lacks ablation experiments. It only presents the performance without using the reward model for data selection. It is suggested to conduct experiments by randomly selecting the same amount of data as picked by the reward model (as random selection is a strong baseline in LLM data selection).
>
> Thank you very much for your suggestion. This experiment is indeed essential. We have added this part to our study, and the results are as follows:
> | Weak Model | Strong Model | Random   | Ours  |
> |------------|--------------|-------|-------|
> | 0.5B       | 1.8B         | 52.49 | 59.17 |
> | 1.8B       | 4B           | 63.52 | 69.05 |
> | 4B         | 7B           | 74.31 | 77.07 |
>
> The results demonstrate that selecting the same amount of data randomly, as opposed to using the reward model, leads to a significant drop in performance. This is primarily due to the reduced number of correctly chosen training samples.
>
> > The legends in the figures are unclear, and some contain labeling errors. In the first three sections of Chapter 3, "weak model" is used as the axis label, yet most of the time, the data shown is based on the performance of the strong model, causing confusion. Additionally, there are errors in the numbering of Figures 3 and 4 in Section 3.3. Besides, there is inconsistent usage of "weak to strong generalization" and "weak-to-strong generalization."
>
> Thank you for your suggestion. We have made the changes in the revision.
>
> > What is the connection between the conclusion regarding the independence of different capabilities mentioned in Weakness 1 and the subsequent methodology and experiments?
>
> Due to the independence of different capabilities, there is no need for additional processing of data selected by the reward model for different capabilities, such as adjusting the data quantity for each capability, in subsequent methodologies and experiments.
>
> > In the scaling experiments, it is observed that as model size increases, the difference between the proposed method and the baseline becomes smaller. Considering the reward model's size is consistent with that of the strong model and the data used for the reward model is from the weak and strong model training, does the proposed method have certain limitations? (In simple terms, is the reward model scalable?)
>
> Firstly, the training data and model size for the reward model are both scalable. In this paper, to maintain the integrity of the weak-to-strong generalization setup and prevent additional factors (e.g., training data and model size) from introducing extra influences, we did not use additional data or larger model sizes.

---

> ### Author Response · Authors · 2024-11-28
> **Official Comment by Authors**
>
> Dear Reviewer r2a7,
>
> We greatly appreciate the time and effort you have taken to review our work. Since the author-reviewer discussion is closing very soon, could you please check our response to see whether it addressed your concerns? We remain open to further discussion and revisions.
>
> Thanks,
>
> The authors

---

> ### Author Response · Authors · 2024-12-04
> **Official Comment by Authors**
>
> Dear Reviewer r2a7,
>
> We greatly appreciate the time and effort you have taken to review our work. We kindly hope that you can take some time to reconsider the rating of our paper based on our responses above. Thank you!
>
> Best regards,
>
> The authors

---

### Official Review · Reviewer_1qFU · 2024-11-08

**Soundness:** 3
**Presentation:** 4
**Contribution:** 2
**Rating:** 6
**Confidence:** 5

**Summary:**

This paper first conducts a series of analyses to explore the factors influencing the spectrum from weak to strong generalization in large language models (LLMs) with multi-capabilities. It then proposes a two-stage training framework that uses reward models to select valuable data based on the insights gained from the analyses. Experimental results demonstrate improvements in the multi-capabilities of the LLMs.

**Strengths:**

- The paper is well-written and easy to understand. Moreover, the structure is complete and coherent.
- Section 3 provides valuable insights, and the proposed method makes sense.

**Weaknesses:**

- The setups of "WTS-S" and "WTS" are not clearly explained upon their first mention. It appears that "WTS-S" involves training different models for different datasets and computing their average score, while "WTS" involves training all datasets together in a single model.
- In Sections 3.3 and 3.4, it would be better to reference the setups under "WTS" rather than "WTS-S".
- What is your setup for the "mix data" in Section 3.6? Are the datasets directly merged? There is concern about potential conflicting cases in the "mix data" that could confuse the model. The total number of correct cases in "mix data" should be larger than "weak data". So I am doubtful about the conclusion regarding Figure 7 (right) because it seems a little contradictory to Figure 7 (left). Additionally, "mix data" should be better termed as "mixed data".
- The performance advantages of "WTS" in Figure 8 are not obvious when the weak model is scaled up. However, the gap to the strong model remains significantly large.
- I doubt the results regarding "the dependence of data quality across various capabilities" as claimed in lines 191-192. In Figure 4, there are only two cases: "w/o GSM8k" and "w/o ECQA & e-SNLI". While these can provide some insight, they are not comprehensive. If, as the conclusion suggests, the abilities in each capability are independent of each other, then why are the mixed training results of multi-capability data better than the single-capability averages?
- **[Important]** Despite the focus on multi-capabilities in the title, I would prefer to see the performance in single capabilities and comparison results with existing methods. It seems that most of the analysis and the proposed method could also apply to single capabilities, which are more commonly used. This paper does not convincingly demonstrate the necessity of working on multiple capabilities.

**Questions:**

- In line 132, there is a missing comma. Additionally, it would be better to explain the difference between your work and related work in the related work section.
- In Figure 1 (right), it would be helpful to annotate that the performance pertains to the strong model.
- In line 186, "Figure 4" should be "Figure 3". In line 191, "Figure 3" should be "Figure 4". Furthermore, in Figures 3 and 4, it would be better to keep the colors of the "weaken" and "remove" bars consistent.
- In Section 3.5, the experiments show that the same strong model cannot improve itself. However, I am interested in the outcomes when training another weak model or a smaller but strong model.

---

> ### Author Response · Authors · 2024-11-25
> **Response to Reviewer 1qFU (1/2)**
>
> Thank you very much for taking the valuable time to review our manuscript and thanks for your positive comments (i.e., valuable insights and the proposed method makes sense). We are sorry for the confusion and unclear expression in the previous version (the main revisions are marked with blue text in the pdf). We have addressed each of the comments and suggestions. Please refer to our responses below for details.
>
> > The setups of "WTS-S" and "WTS" are not clearly explained upon their first mention. It appears that "WTS-S" involves training different models for different datasets and computing their average score, while "WTS" involves training all datasets together in a single model. In Sections 3.3 and 3.4, it would be better to reference the setups under "WTS" rather than "WTS-S".
>
> Thank you for your suggestion. We have already incorporated the changes in the revision.
>
> > What is your setup for the "mix data" in Section 3.6? Are the datasets directly merged? There is concern about potential conflicting cases in the "mix data" that could confuse the model. The total number of correct cases in "mix data" should be larger than "weak data". So I am doubtful about the conclusion regarding Figure 7 (right) because it seems a little contradictory to Figure 7 (left). Additionally, "mix data" should be better termed as "mixed data".
>
> This misunderstanding arose due to our unclear explanation, and we have corrected it in the revision. Thank you for pointing out the confusion.
> First, "mix data" is unrelated to Figure 7 (left).
> In Figure 7 (left), we split the dataset generated by the weak model into two parts: correct samples (clean data) and incorrect samples (noisy data). We then progressively add noisy samples to the clean data at different ratios to train the strong model.
> In Figure 7 (right), we directly combine the datasets generated by the strong and weak models and train the model for half the original number of epochs. Therefore, the conclusions drawn from Figure 7 (left) and Figure 7 (right) are not in conflict. As observed in Figure 5, the dataset labeled by the strong model (strong dataset) is of lower quality. Thus, training on the mixed dataset leads to better performance compared to training solely on the strong dataset, which is a reasonable outcome.
>
> > The performance advantages of "WTS" in Figure 8 are not obvious when the weak model is scaled up. However, the gap to the strong model remains significantly large.
>
> First, we can confirm that the performance of "WTS" cannot surpass that of a "strong model" trained solely on clean samples. From Figure 9, the relatively low "Performance Gap Recovered" aligns with your observation that the models exhibit poor recovery in weak-to-strong generalization from 4B to 7B. However, the primary reason for this lies in the relatively small difference in parameter size between the 4B and 7B models. As shown in Figure 14, when using models with significantly larger parameter sizes, we observe a notable improvement in the "Performance Gap Recovered."
>
> > I doubt the results regarding "the dependence of data quality across various capabilities" as claimed in lines 191-192. In Figure 4, there are only two cases: "w/o GSM8k" and "w/o ECQA & e-SNLI". While these can provide some insight, they are not comprehensive. If, as the conclusion suggests, the abilities in each capability are independent of each other, then why are the mixed training results of multi-capability data better than the single-capability averages?
>
> There seems to be a misunderstanding that needs to be clarified. In our paper, we did not claim that multi-capability weak-to-strong generalization performs better than single-capability weak-to-strong generalization. As stated in our conclusion, the capabilities within each category are independent of each other. The single-capability averages approximate the WTS performance, as shown below:
> | Weak Model | Strong Model | Ours (single-capability averages) | Ours  |
> |------------|--------------|------------------------------------|-------|
> | 0.5B       | 1.8B         | 59.18                             | 59.17 |
> | 1.8B       | 4B           | 69.04                             | 69.05 |
> | 4B         | 7B           | 77.06                             | 77.07 |

---

> ### Author Response · Authors · 2024-11-25
> **Response to Reviewer 1qFU (2/2)**
>
> > [Important] Despite the focus on multi-capabilities in the title, I would prefer to see the performance in single capabilities and comparison results with existing methods. It seems that most of the analysis and the proposed method could also apply to single capabilities, which are more commonly used. This paper does not convincingly demonstrate the necessity of working on multiple capabilities.
>
> First, as clarified in the previous discussion, our method is applicable for single-capability weak-to-strong generalization. We also provide a performance comparison with the state-of-the-art Weak-to-Strong Generalization method (CSL) [1], as shown below:
>
> | Weak Model | Strong Model | CSL   | Ours  |
> |------------|--------------|-------|-------|
> | 0.5B       | 1.8B         | 57.84 | 59.17 |
> | 1.8B       | 4B           | 68.62 | 69.05 |
> | 4B         | 7B           | 76.91 | 77.07 |
>
> The results clearly demonstrate the advantages of our approach.
> In addition, weak-to-strong generalization can be applied to large-scale models with significant parameter counts. Compared to single-capability strong models, we may place greater emphasis on multi-capability large models like GPT-4 and Gemini, which are more practical in real-world scenarios. Furthermore, existing models optimized for specialized capabilities also tend to consider their performance in other capabilities [2].
>
> [1] Co-Supervised Learning: Improving Weak-to-Strong Generalization with Hierarchical Mixture of Experts\
> [2] TIMO: Towards Better Temporal Reasoning for Language Models
>
>
> > In line 132, there is a missing comma. Additionally, it would be better to explain the difference between your work and related work in the related work section. In Figure 1 (right), it would be helpful to annotate that the performance pertains to the strong model. In line 186, "Figure 4" should be "Figure 3". In line 191, "Figure 3" should be "Figure 4". Furthermore, in Figures 3 and 4, it would be better to keep the colors of the "weaken" and "remove" bars consistent.
>
> Thank you for your suggestion. We have made the changes in the revision.
>
> > In Section 3.5, the experiments show that the same strong model cannot improve itself. However, I am interested in the outcomes when training another weak model or a smaller but strong model.
>
> We believe your perspective is promising. But, a smaller model is not necessarily a weaker model, as it can benefit from longer training times and better-quality data. Therefore, we do not explore this aspect in this work to avoid introducing too many variables.

---

> ### Author Response · Authors · 2024-11-28
> **Official Comment by Authors**
>
> Dear Reviewer 1qFU,
>
> We greatly appreciate the time and effort you have taken to review our work. Since the author-reviewer discussion is closing very soon, could you please check our response to see whether it addressed your concerns? We remain open to further discussion and revisions.
>
> Thanks,
>
> The authors

---

> > ### Comment · Reviewer_1qFU · 2024-12-03
> >
> > Thanks for your reply. I have raised my score from 5 to 6. I hope the final version can make the motivation of multi-capability and some details more clearly and add the comparison with single-capability weak-to-strong generalization.

---

> > > ### Author Response · Authors · 2024-12-03
> > > **Official Comment by Authors**
> > >
> > > Thank you very much for the score improvement and your constructive feedback. In the final version, we will further clarify the motivation for multi-capabilities and details and add a comparison with single-capability weak-to-strong generalization. Thank you!

---

### Author Response · Authors · 2024-11-27
**Request for Feedback Before Discussion Deadline**

Dear Reviewers and Area Chair,

As the discussion deadline approaches, we kindly request all reviewers to review our responses to your comments and share any remaining concerns or questions. We highly value your feedback and want to ensure that all issues are fully addressed before the final decision.

We have carefully responded to all of the concerns raised during the review and discussion phases by adding additional experiments and explanations. We hope that these clarifications might lead to a reconsideration of the current score.

We sincerely appreciate your valuable time and effort in reviewing our submission.

Best regards,

The authors

---

### Author Response · Authors · 2024-12-02
**Request for Feedback Before Extended Discussion Deadline**

Dear Reviewers and Area Chair,

As the extended discussion deadline is fast approaching, we kindly request all reviewers to provide feedback regarding our responses to your comments. We have carefully addressed all concerns raised in the review, including conducting additional experiments and providing more detailed explanations. We are hopeful that these updates will prompt a reconsideration of the current rating. Your feedback is immensely valuable to us.

Thank you again for your time and effort in reviewing our work.

Best regards,

The Authors

---

### Author Response · Authors · 2024-12-03
**General Response to Reviewers and Revision Submitted**

Dear all reviewers and ACs,

Thank you very much for your valuable time in providing these constructive comments and suggestions. We have addressed each of the comments and misunderstanding by adding more experiments or explanations.

### We also thank the reviewers for identifying the strengths of our work below:
1. Reviewer 1qFU highlights Section 3 (analysis of multi-capabilities weak to strong generalization) in our study provides valuable insights, and the proposed method makes sense.
2. Reviewer r2a7 appreciates our work analyzing the current gaps in research on the weak to strong generalization paradigm from a "super-alignment" perspective and providing a novel training approach under this setup.
3. Reviewer 5nVp highlights our work provides relevant conclusions for multi-capabilities weak-to-strong generalization and explores data selection directions for weak-to-strong generalization.
4. Reviewer fdJr praises our research for its novelty and significance, highlighting valuable insights into multi-capability alignment under super-alignment. The writing is commended for its rigor and depth, and the proposed two-stage approach is recognized for effectively addressing key challenges, supported by solid theory and experimental validation.
5. Reviewer orGJ appreciates our insight into how weak supervision affects strong models, revealing their limitations. The proposed framework, using reward models to select valuable data, is innovative and addresses overconfidence while improving generalization. The paper is clearly presented and easy to understand.


### We have revised the paper to address the reviewers’ concerns. Below we summarize the major revisions (the main revisions are marked with blue text in the pdf, we also made some minor layout changes to fit the page limit), while we reply to the comments of each reviewer separately. The major revisions are:


1. We have further clarified the content in the figures, the details of the multi-capability setting, and the experimental setup (Reviewers 1qFU, r2a7, 5nVp, fdJr, orGJ).
2. We have added ablation experiments (randomly selecting datasets) to verify the effectiveness of our reward model (Reviewer r2a7).
3. Our method can also be applied to single-capability weak-to-strong generalization, achieving results comparable to those in the multi-capability setting. Moreover, it outperforms the state-of-the-art weak-to-strong generalization method (CSL) [1]. We hope these results address the concerns raised by Reviewer 1qFU.
4. We compared our method with an untrained strong model with in-context learning, demonstrating the significance of studying weak-to-strong generalization (Reviewer 5nVp).

We report a simplified version of some of the added results in the table below, and the detailed complete results are in Appendix of the paper.

1. Our Method for Single-Capability and Multi-Capabilities Weak to Strong Generalization
| Weak Model | Strong Model | Ours (single-capability averages) | Ours |
|------------|--------------|-----------------------------------|------|
| 0.5B       | 1.8B         | 59.18                             | 59.17 |
| 1.8B       | 4B           | 69.04                             | 69.05 |
| 4B         | 7B           | 77.06                             | 77.07 |


2. Comparison with SOTA (CSL) [1]
| Weak Model | Strong Model | CSL  | Ours |
|------------|--------------|------|------|
| 0.5B       | 1.8B         | 57.84| 59.17 |
| 1.8B       | 4B           | 68.62| 69.05 |
| 4B         | 7B           | 76.91| 77.07 |

3. Ablation Study
| Weak Model | Strong Model | Random | Ours |
|------------|--------------|--------|------|
| 0.5B       | 1.8B         | 52.49  | 59.17 |
| 1.8B       | 4B           | 63.52  | 69.05 |
| 4B         | 7B           | 74.31  | 77.07 |


4. Comparison with Untrained Strong Model (w/ In-Context Learning)
| Weak Model | Strong Model | Ours  | ICL   |
|------------|--------------|-------|-------|
| 0.5B       | 1.8B         | 59.19 | 45.92 |
| 1.8B       | 4B           | 69.07 | 58.38 |
| 4B         | 7B           | 77.06 | 70.02 |

[1] Co-Supervised Learning: Improving Weak-to-Strong Generalization with Hierarchical Mixture of Experts

---

### Meta-Review · Area_Chair_DKCv · 2024-12-20

**Metareview:**

This paper addresses the superalignment challenge in LLMs by exploring the weak to strong generalization with multi-capabilities. The authors propose a novel training framework using reward models to select valuable data for weak supervision and introduce a two-stage training method that enhances model performance. The paper is well-written, providing valuable insights into the multi-capability generalization and presenting experimental results that demonstrate improvements over baseline methods.

The reviewers raised some questions about the clarity of certain setups, such as the "WTS-S" and "WTS" configurations, and the handling of mixed data which could potentially introduce conflicting cases. There were also concerns about the necessity of focusing on multi-capabilities when the results could potentially apply to single capabilities as well. During the rebuttal, the authors addressed these issues by clarifying their experimental setups and providing additional justifications for the focus on multi-capabilities. The reviewers were generally satisfied with the authors' responses.

**Additional Comments On Reviewer Discussion:**

Nil

---

### Decision · Program_Chairs · 2025-01-22

Accept (Poster)